# Sidechain conditioning and modeling for full-atom protein sequence design with FAMPNN

Talal Widatalla [* 1 2]   Richard W. Shuai [* 1]   Brian L. Hie [† 2 3 4]   Po-Ssu Huang [† 5]

## Abstract

Leading deep learning-based methods for fixed-backbone protein sequence design do not model protein sidechain conformation during sequence generation despite the large role the three-dimensional arrangement of sidechain atoms play in protein conformation, stability, and overall protein function. Instead, these models implicitly reason about crucial sidechain interactions based on backbone geometry and known amino acid sequence labels. To address this, we present FAMPNN (Full-Atom MPNN), a sequence design method that explicitly models both sequence identity and sidechain conformation for each residue, where the per-token distribution of a residue's discrete amino acid identity and its continuous sidechain conformation are learned with a combined categorical cross-entropy and diffusion loss objective. We demonstrate that learning these distributions jointly is a highly synergistic task that both improves sequence recovery while achieving state-of-the-art sidechain packing. Furthermore, benefits from full-atom modeling generalize from sequence recovery to practical protein design applications, such as zero-shot prediction of experimental binding and stability measurements.

## 1. Introduction

Most existing methods for designing protein sequences for a given structure do not explicitly model or encode sidechains. Instead, they formulate the sequence design task as a backbone-conditioned sequence generation task. These models typically use a graph neural network (GNN) to encode the backbone structure, then generate the sequence either autoregressively or through a masked language modeling objective (Dauparas et al., 2022; Hsu et al., 2022; Ingraham et al., 2019; Ruffolo et al., 2024; Zheng et al., 2023; Gao et al., 2022a; Akpinaroglu et al.). Because these methods do not model sidechains during sequence design, they must implicitly reason about sidechain interactions through the limited perspective of backbone structure and sequence identity alone.

We summarize our key contributions as follows:

- We introduce a method which models the per-token distribution of a residue's discrete sequence identity and continuous sidechain structure with a combined cross-entropy and diffusion loss objective.
- We implement a lightweight, iterative sampling method for generating samples from this joint distribution.
- We demonstrate that modeling this joint distribution allows for improved sequence design and experimental protein fitness prediction.

## 2. Related work

### 2.1. Fixed-backbone sequence design

The task of fixed-backbone sequence design is to design a protein sequence that will fold into a structure that matches the given protein backbone (Dauparas et al., 2022). Traditional physics-based approaches use energy functions to combinatorially optimize sequence identity and sidechain conformation to find low-energy configurations (Holm & Sander, 1992; Dahiyat & Mayo, 1997).

Recently, deep learning-based approaches have proven highly successful at this task (Jing et al., 2020; Norn et al., 2021; Hsu et al., 2022; Gao et al., 2022b;a; Zheng et al., 2023; Anand et al., 2022; Akpinaroglu et al.; Ruffolo et al., 2024; Ingraham et al., 2023). In addition to being able to recover the sequence of natural proteins given their backbone, these models are surprisingly capable of also guiding improvements in binding affinity (Shanker et al., 2024) and improving protein expression, solubility, and stability (Sum-

---

[*]Equal contribution. Either may be listed first on a CV. [†]Co-corresponding authors. [1]Department of Biophysics, Stanford University, Stanford, CA [2]Arc Insitute, Palo Alto, CA [3]Department of Chemical Engineering, Stanford University, Stanford, CA [4]Stanford Data Science, Stanford University, Stanford, CA [5]Department of Bioengineering, Stanford University, Stanford, CA. Correspondence to: Po-Ssu Huang <possu@stanford.edu>, Brian L. Hie <brianhie@stanford.edu>.

*Proceedings of the 42$^{nd}$ International Conference on Machine Learning*, Vancouver, Canada. PMLR 267, 2025. Copyright 2025 by the author(s).

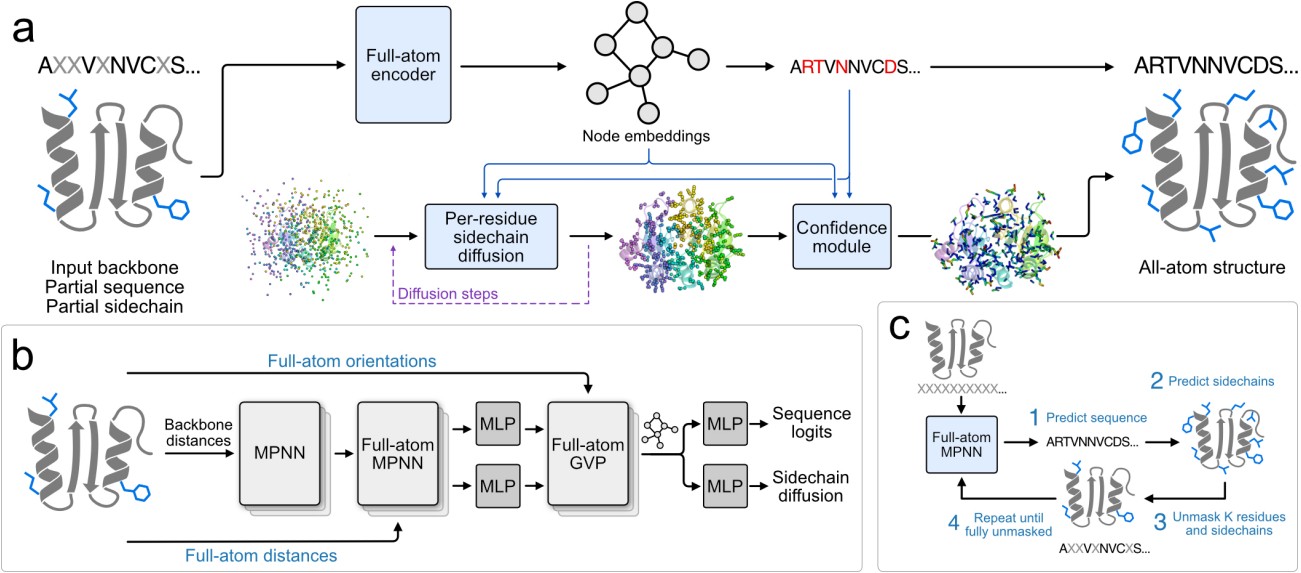

*Figure 1.* **(a)** Full-atom sequence design overview, consisting of a dual masked language modeling scheme on sequence and sidechain structure to co-generate sequence and sidechain atoms. **(b)** Given a backbone with partially masked sequence and sidechains, the full-atom encoder uses modified MPNN and GVP layers to produce node embeddings, which are used for sequence prediction and sidechain generation. **(c)** Sampling begins with a fully-masked backbone and is iteratively unmasked to generate sequence and sidechains.

ida et al., 2024). Among these, ProteinMPNN currently remains the most widely used model for sequence design due to its proven experimental performance for various practical protein design applications (Dauparas et al., 2022; Bennett et al., 2023; Dauparas et al., 2023).

## 2.2. Sidechain packing

The task of sidechain packing is, given a protein backbone and its amino acid sequence, to predict the sidechain coordinates of the ground truth structure. Traditional physics-based approaches for sidechain packing minimize energy by sampling rotamers from a rotamer library (Krivov et al., 2009; Xu & Berger, 2006; Miao et al., 2011). Following the groundbreaking success of deep learning approaches in protein structure prediction (Abramson et al., 2024; Jumper et al., 2021), recent specialized sidechain packing methods have achieved strong performance over physics-based methods. DLPacker formulates the problem as an image-to-image task and trains a 3D U-Net to predict sidechain coordinates (Misiura et al., 2022). AttnPacker uses an invariant graph transformer with triangle updates followed by an equivariant Tensor Field Network Transformer to predict sidechain coordinates (McPartlon & Xu, 2023). PIPPack introduces invariant point message passing layers to predict binned $\chi$-angle distributions (Randolph & Kuhlman, 2023). DiffPack uses torsional diffusion to autoregressively generate the four sidechain torsional angles $\chi_1$ to $\chi_4$ (Zhang et al., 2024). FlowPacker uses torsional flow matching with an equivariant graph neural network to simultaneously denoise all four sidechain torsional angles (Lee & Kim, 2024).

## 2.3. Sequence-sidechain co-generation methods

Closest to our work are models that perform both sequence design and sidechain packing. AttnPacker is trained with an auxiliary task of simultaneously predicting amino acid sequence identity by randomly dropping out residue identities in the input sequence. However, their work primarily focuses on use cases for sidechain packing rather than sequence design. The 3DCNN model from Anand et al. can sample sequence and sidechains conditioned on neighboring sidechains (Anand et al., 2022), but it requires a computationally expensive Markov Chain Monte Carlo (MCMC) procedure to iteratively redesign residues. The high demand for computational resources reduces its broad application. LigandMPNN extends ProteinMPNN to design sequences conditioned on ligand context, which can optionally include sidechain atoms as ligand context, but it remains unable to co-generate sidechains and sequence (Dauparas et al., 2023). Instead, LigandMPNN packs sidechains by relying on a separately trained model after the full sequence has been designed, meaning that the sequence is designed completely independently of predicted neighboring sidechain conformations.

## 2.4. Next-token modeling for continuous data

Autoregressive models for generating continuous data often rely on discretizing the data before tokenization (Ramesh et al., 2021; Esser et al., 2021; Chang et al., 2022). To eliminate the need for discrete tokenizers, recent approaches have used a diffusion loss objective with autoregressive

modeling to model per-token distributions for continuous-valued data (Li et al., 2024; Fan et al., 2024; Sun et al., 2024). Our work similarly uses a diffusion loss to extend masked generative models to generate tokens with both discrete and continuous components.

## 3. Motivation

The structure and function of proteins are largely determined by the physicochemical interactions among its atoms and its system. Current fixed-backbone sequence design methods implicitly capture these interactions via backbone geometry and amino acid identity labels alone, and the resulting sequences often achieve remarkable solubility and stability. However, these methods fall short on tasks involving nuanced sidechain arrangements. There are several inter-molecular interactions which are functions of the precise distances (e.g. Van der Waals force and Coulombic attraction) and angles (e.g. hydrogen bonding and pi-stacking), amongst sidechain and backbone atoms. Thus, we hypothesize that by explicitly modeling full-atom structure in an efficient manner during sequence design, we can better capture complex, yet crucial, sidechain interactions unavailable to backbone methods, improving the quality of designed sequences.

To carry out this hypothesis, our design process aims to achieve two goals:

1. At each step in the sequence design process, the model should co-generate sidechain atoms along with its prediction of the sequence.

2. To predict sequence and sidechains, the model should encode the currently known sidechain atoms in addition to the backbone and sequence.

## 4. Methods

### 4.1. Generative masked language modeling scheme

We train FAMPNN with masked language modeling on sequence identity. Simultaneously, we provide sidechain coordinates of unmasked sequence as context, and mask the sidechain coordinates of masked sequence positions. As seen in Figure **1a**, this allows for prediction of masked sequence and sidechain conformations given partial sequence and sidechain coordinates. To predict both sequence and sidechains, during training, we combine categorical cross-entropy loss for sequence prediction and per-token diffusion loss for sidechain conformation prediction, such that FAMPNN is a single model trained end-to-end on both objectives (Appendix **C**).

Because our method generates both sequence and sidechain coordinates in a per-token manner, we can formulate the full-atom sequence design procedure as a generative masked language modeling objective. Similar to MaskGIT, (Chang et al., 2022) we begin sampling with both sequence and sidechains fully masked and follow an iterative procedure: at each step, we predict all tokens in parallel, then unmask some subset of the sequence and sidechain tokens. We repeat this process until all tokens are unmasked, yielding a fully designed sequence and full-atom structure (Figure **1c**). Sampling with as many steps as residues reduces to random-order autoregression.

To additionally support sidechain packing, a crucial task in protein design, we introduce a separate masking scheme exclusively on sidechains. During training, among positions with unmasked sequence identity, we randomly mask each residue's sidechain coordinates with probability $p \sim \text{Uniform}(0,1)$. This enables the model to predict sidechain conformations given either partial or complete sequence information, with varying amounts of known sidechain information. This also allows for the model to predict sequence identity given a mixture of sequence-only and sequence-and-sidechain context.

### 4.2. Full-atom sequence design

Fulfilling the sequence design process while explicitly modeling sidechain structure requires addressing two primary considerations. (1) Determining a protein sequence given its full-atom structure is trivial, as the composition of sidechain atoms completely identifies the underlying amino acids, and (2), sidechains for different amino acids have a varying number of atoms, so we require a unifying representation for all sidechains.

#### 4.2.1. SIDECHAIN COORDINATE REPRESENTATION

To address these considerations, we represent residues in `atom37` format, in which each residue is represented as a fixed size matrix of size $37 \times 3$. Each row corresponds to the 3D coordinates of all 37 possible atom types. 4 of these rows correspond to the backbone atoms N, C$\alpha$, C, and O, while the remaining 33 rows correspond to the residue's sidechain atoms. For sidechains where a particular atom type is not present, the row for that atom type is a "ghost atom" and is set to the residue's C$\alpha$ position, accommodating consideration (2). Utilization of ghost atoms also allows us to address consideration (1) as we can prevent sidechain structure from revealing the amino-acid identity of a masked sequence position by setting all sidechain atoms as ghost atoms, essentially "masking" the sidechain structure as well.

#### 4.2.2. FULL-ATOM ENCODER

We represent full-atom protein structure as a graph, encoded with a graph neural network (GNN) using a hybrid MPNN-GVP architecture (Dauparas et al., 2022; Jing et al.,

2020). This architecture consists of three primary components, which are an invariant backbone encoder, an invariant full-atom encoder, and an equivariant full-atom encoder, with respect to any composition of global rotations and reflections in protein coordinates.

The first two components build off the architecture of ProteinMPNN (Dauparas et al., 2022). ProteinMPNN featurizes backbone structure as a k-NN graph, with a set of nodes $\mathcal{V}$ that represent each protein residue, and edges $\mathcal{E} = \{e_{i \to j}\}$, featurized with backbone inter-atomic distances, defined for the 48 nearest neighbors of each residue. ProteinMPNN includes an initial structure encoder consisting of three previously described (Ingraham et al., 2019) invariant MPNN layers, which update both edge representations $e_{ij}$ and node representations $v_i$: MPNNEncoder($v_i, e_{ij}$) = ($v_i', e_{ij}'$). This output is passed to a sequence decoder comprised of three additional invariant MPNN layers that combines causally-masked one-hot encodings of sequence $\mathcal{S}$, updating only node representations: MPNNDecoder($v_i', [e_{ij}', s_i]$) = ($v_i''$). This output is then passed into a final output head for sequence prediction.

Our initial component, the invariant backbone encoder, is identical to MPNNEncoder, encoding the backbone structure only. However, for our second component, we replace MPNNDecoder with a full-atom encoder, which is identical to the backbone encoder, MPNNEncoder, but with expanded featurization to all atoms. Similar to MPNNDecoder, we concatenate one-hot encodings of sequence identity, but remove the causal mask, as FAMPNN is trained with an MLM objective, and sequence positions are randomly masked with a corresponding token.

Lastly, to allow the model to reason over vector-valued interatomic orientations in addition to previously encoded scalar-valued interatomic distances, we require equivariant layers, since atomic orientations are equivariant to global rotations and reflections. Thus, the third and final component of the model are modified Geometric Vector Perceptrons (GVP) from Jing et al. (Jing et al., 2020). GVP layers consist of an equivariant track for learning vector features and an invariant track for learning scalar features. In FAMPNN, in addition to the backbone-level features used in Jing et al. (Jing et al., 2020), we use the equivariant track to encode unit vectors from $C_{\alpha_i}$ to all other atoms in residue $i$. For equivariant edge features, we include unit vectors from $\mathbf{C}_{\alpha_i}$ to all atoms in residue $j$. To the invariant track, we additionally incorporate distances from $\mathbf{C}_{\alpha_i}$ to all atoms in residue $j$, as done in the invariant full-atom encoder.

To tie the three modules together, we initialize the invariant track of GVP with the final node and edge representations from the invariant full-atom encoder. This representation is then passed both to a final output head for sequence prediction and to the sidechain diffusion module for invariant Euclidean denoising of sidechain atoms.

## 4.3. Sidechain coordinate generation

We formulate sidechain coordinate generation as per-token Euclidean diffusion with a diffusion loss (Li et al., 2024) on coordinates predicted from the node embeddings. During sidechain coordinate generation, we use a sidechain representation that is invariant to global rotations and translations of the input backbone, which we have empirically seen improves training efficiency. For each residue, we transform all sidechain coordinates (including ghost atoms) into the local frame defined by its backbonande atoms (Appendix **D.3.1**). At inference, after a full denoising trajectory has been run, sidechain coordinates are transformed back into the global frame by inverting this transformation.

### 4.3.1. PER-TOKEN EUCLIDEAN DIFFUSION

We use diffusion to model the distribution of a residue's sidechain coordinates given its amino acid identity, the protein backbone, and the currently unmasked sequence and sidechains. Similar to Chu et al., our diffusion scheme follows the variance-exploding EDM scheme from Karras et al. (Karras et al., 2022; Chu et al., 2024), where we train a noise-conditioned denoiser $D_\theta$ to minimize the $L_2$ error from a Gaussian-noised version of the sidechain coordinates. We parametrize $D_\theta$ using a lightweight multilayer perceptron (MLP). We use AdaLN (Peebles & Xie, 2023) to condition this diffusion MLP on the current noise level $\sigma$ and a one-hot encoding of the residue's amino acid identity $s_i$. To encode information about the backbone and unmasked sequence and sidechains, we also use AdaLN to pass in the node embedding $v_i$ from our full-atom encoder.

At train time, when denoising the sidechain coordinates of residue $i$, we teacher-force the sequence $s_i$ rather than predicting $s_i$. Similar to previous work (Abramson et al., 2024; Li et al., 2024), because the diffusion MLP is lightweight relative to the rest of the network, we train the MLP with a larger batch size for better training efficiency. In practice, for each training example, we clone the conditioning inputs to the MLP 8 times and sample 8 different noise levels for training the denoiser. At inference time, to generate sidechain atoms for residue $i$, we first run the MPNN module to generate a prediction of the sequence identity $\hat{s}_i$, then pass in both $\hat{s}_i$ and $v_i$ into the sidechain diffusion MLP. We then initialize all sidechain atoms from a Gaussian distribution and run a diffusion trajectory to realize sidechain coordinates (Figure **1a**).

### 4.3.2. PREDICTED SIDECHAIN ERROR

Predicted error in protein structure prediction models is valuable for interpretability, design filtering, and as an optimization objective for *de novo* design (Abramson et al.,

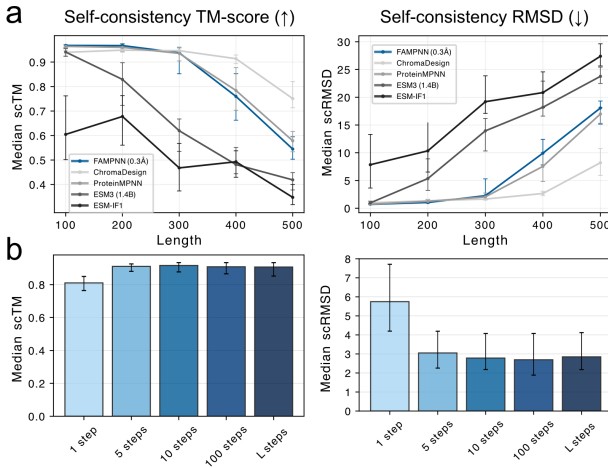

*Figure 2.* **(a)** Median self-consistency TM-scores (left, higher is better) and RMSDs (right, lower is better) for sequence design methods across 100 *de novo* backbones at each length in {100, 200, 300, 400, 500} generated by RFdiffusion. **(b)** Median self-consistency TM-scores (left) and RMSDs (right) across all 500 *de novo* backbones vs. number of FAMPNN iterative unmasking steps. L steps denotes unmasking 1 residue per step, which is equivalent to random order autoregression. Error bars represent 95% bootstrap confidence intervals around the median.

*Table 1.* Median sequence accuracy on CATH 4.2 test set, as reported by Gao et. al (Gao et al., 2022a). Bold values indicate the best results. †Evaluated on CATH 4.3 test set (Hsu et al., 2022). *Reproduced by Gao et. al with CATH 4.2 (Gao et al., 2022a). Dashes (-) indicate that perplexity was not reported by the authors.

| Model | Recovery % ↑ | Perplexity ↓ |
|---|---|---|
| GraphTrans | 35.82 | 6.63 |
| StructGNN | 35.91 | 6.40 |
| GCA | 37.64 | 6.05 |
| ESM-IF1† | 38.30 | 6.44 |
| GVP-large† | 39.20 | 6.17 |
| GVP | 39.47 | 5.36 |
| AlphaDesign | 41.31 | 6.30 |
| ProteinMPNN* | 45.96 | 4.61 |
| ESM-IF1 (AF2DB)† | 51.60 | **4.01** |
| PiFold | **51.66** | 4.55 |
| Frame2Seq | 46.53 | - |
| Frame2Seq (Ensemble) | 49.11 | - |
| FAMPNN | 49.66 | 4.99 |

2024; Frank et al., 2024; Bryant et al., 2022). Similar to other sidechain packing methods, we train a confidence module to predict packing error (Zhang et al., 2024; Lee & Kim, 2024; McPartlon & Xu, 2023). Our confidence module estimates predicted sidechain error (pSCE) as a per-atom error in Angstroms (Figure **1**). Similar to AlphaFold3, during confidence module training, we use a diffusion rollout to obtain sidechain coordinates for computing error against the ground truth (Abramson et al., 2024). The confidence module encodes node embeddings from the full-atom encoder with 3 MPNNDecoder layers, then uses the predicted sidechain coordinates and amino acid type to predict the per-atom error in evenly spaced bins from 0Å to 4Å. At inference time, we compute pSCE by taking the expectation across these binned probabilities (Appendix **D.4**).

## 5. Results

### 5.1. Full-atom sequence design

#### 5.1.1. SEQUENCE RECOVERY AND SELF-CONSISTENCY

To benchmark FAMPNN's sequence recovery compared to other methods, we report median sequence recovery on the CATH 4.2 test set in Table **1**. FAMPNN (0.0Å) achieves a competitive 49.66% single-step sequence recovery, surpassing the two modules, ProteinMPNN (45.96%) and GVP (39.47%), which make up the hybrid MPNN-GVP architecture. We omit benchmarking of protein design models

which utilize pre-trained protein language models such as ESM2 (Lin et al., 2023), as the pre-trained modules do not explicitly holdout the CATH 4.2 test set.

Because high sequence recovery does not always translate to high protein design success rates, another critical metric for evaluating protein design methods is self-consistency, a measure of how well the designed sequence is predicted to recapitulate the input backbone. Specifically, we compute the TM-score (structural similarity) and RMSD of the input structure and the AlphaFold2 structure prediction of the designed sequence, reported as scTM and scRMSD respectively (Zhang & Skolnick, 2005). Similar to the ProteinBench inverse folding benchmark, we evaluate on RFdiffusion-generated *de novo* backbones from lengths 100 to 500 (Ye et al., 2024a). For each length in {100, 200, 300, 400, 500}, we evaluate various sequence design methods on 100 *de novo* backbones. Shown in Figure **2a**, FAMPNN (0.3Å) is competitive with commonly used ProteinMPNN in scTM and scRMSD. For a comparison of all benchmarked methods, see Appendix Table **6**. In Figure **2b**, we show the effect of the number of iterative FAMPNN sampling steps on self-consistency. Performing only 1 step of sequence design is insufficient to capture higher-order sequence interactions, resulting in worse self-consistency, but we find that 10 steps is sufficient to achieve high self-consistency. Using a smaller number of iterative steps significantly reduces inference time compared to a fully autoregressive approach of L steps (e.g. ProteinMPNN and ESM-IF1), which requires one step per residue.

*Table 2.* Sidechain packing performance comparison across CASP datasets. Bold values indicate the best results. Underlined values indicate second best. Methods marked with an asterisk (*) were trained on a dataset that does not hold out CASP proteins.

| Dataset | Method | Atom RMSD (Å) ↓ All / Core / Surface |
|---------|--------|------------------------------------|
| CASP15 | 3DCNN* | 0.897 / 0.489 / 1.003 |
|  | ChromaDesign* | 0.810 / 0.434 / 0.917 |
|  | LigandMPNN* | 0.788 / 0.443 / 0.879 |
|  | FlowPacker | 0.765 / 0.452 / 0.859 |
|  | FAMPNN (0.3Å) | 0.785 / 0.417 / 0.888 |
|  | FAMPNN (0.0Å) | **0.690** / **0.350** / **0.789** |

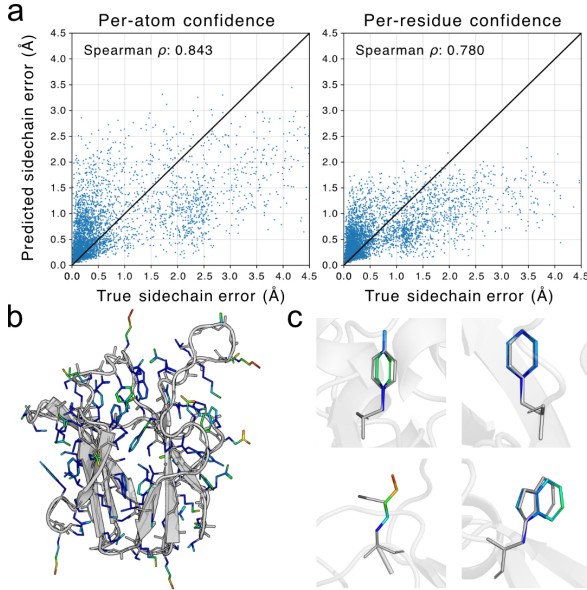

## 5.2. Sidechain packing

Following FlowPacker (Lee & Kim, 2024), we compare our sidechain packing performance to other methods on CASP13, 14, and 15 targets (Table **2**). Following Lee and Kim, we used MMseqs2 `easy-search` to remove all CASP13-15 homologues from our training and validation dataset with a cutoff of 40% similarity. We measure sidechain packing performance by the average RMSD between the predicted and ground-truth sidechains. Unlike most methods that predict sidechain coordinates via torsion angles (Anand et al., 2022; Zhang et al., 2024; Dauparas et al., 2023; Lee & Kim, 2024), our method uses an MSE-based objective in Cartesian space. This avoids the lever arm effect, where small angular errors cause large Cartesian deviations, and achieves superior RMSDs while remaining competitive on chi-angle accuracy (Appendix Table **8**).

We also demonstrate that pSCE serves as an effective confidence metric for sidechain packing. We evaluated pSCE on crystal structures from the CASP15 test set and found strong correlations in both per-atom errors (Spearman $\rho = 0.843$) and per-residue errors (Spearman $\rho = 0.780$), as shown in Figure **3a**. Figure **3b** displays a visualization of confidence on a packed structure, where confidence can be meaningfully interpreted on a per-atom level. For example, we observe that sometimes, while the hydroxyl tip of a tyrosine sidechain is predicted with high confidence, the model assigns lower confidence to the ring's rotational orientation (Figure **3c**, top left).

Additionally, pSCE remains interpretable for designed sequences predicted to fold close to the target backbone (scRMSD $\leq$ 5Å). We find that pSCE correlates well with the RMSD between FAMPNN-packed sidechains and AlphaFold2 predictions, when sidechains are aligned to the backbone reference frame, suggesting its utility as a confidence metric for sequence design (Figure **4**).

*Figure 3.* **(a)** Predicted sidechain error correlates with true sidechain packing error both per-atom (left) and per-residue by averaging over atoms (right). **(b)** Visualization of a repacked target from CASP15 (ID: T1145-D1). Sidechain color represents confidence, with blue representing low predicted error and red representing high predicted error. **(c)** Specific FAMPNN-packed sidechains with per-atom confidence (colored), along with the ground truth conformation (gray).

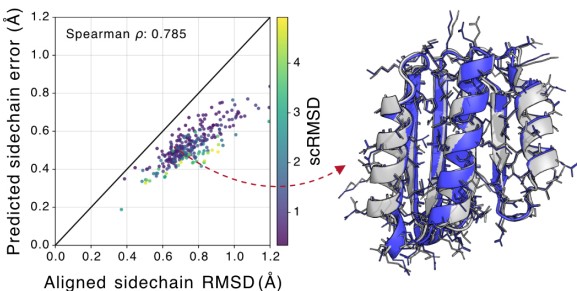

*Figure 4.* Left, among designed sequences achieving scRMSD $\leq$ 5Å, our predicted sidechain error correlates well with the actual RMSD between FAMPNN-packed sidechains and those predicted by AlphaFold2, when each sidechain is aligned in its backbone frame. Right, AlphaFold2-predictions (blue) of designed sequences can closely match the target backbone (gray), with predicted sidechains closely matching FAMPNN-designed sidechains.

## 5.3. Full-atom conditioned protein fitness evaluation

Current inverse folding, or structure-conditioned sequence design models, have a surprising capacity to score mutant protein fitness zero-shot, without observing any fitness properties during pretraining. Markedly, protein stability and protein-protein binding affinity are two properties where inverse folding models significantly outperform protein language models (Shanker et al., 2024; Notin et al., 2023;

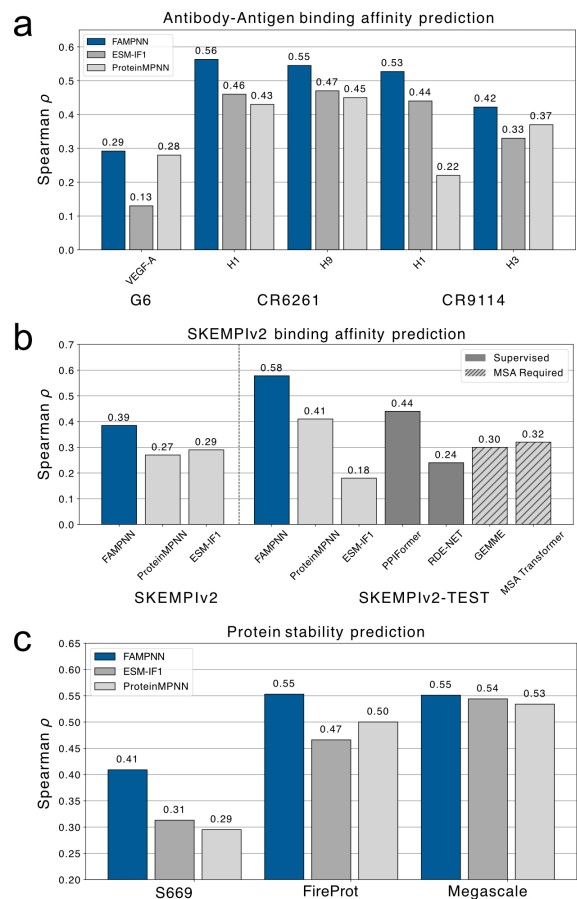

*Figure 5.* Zero-shot performance of FAMPNN against state-of-the-art protein design methods in prediction of experimentally measured protein fitness. **(a)** Prediction of $\Delta\Delta G$ of binding of protein variants in the SKEMPIv2 dataset, reported with Spearman correlation. We additionally evaluate on a separate test subset of SKEMPIv2 recently curated to address significant data leakage issues in supervised models trained on previously proposed data splits (Bushuiev et al., 2024). **(b)** Prediction of antibody-antigen binding affinity evaluated on five antibody-antigen binding affinity datasets (Shanker et al., 2024), reported with Spearman correlation. Antibody is shown in bold below the plot, and corresponding antigen target shown on x-axis. **(c)** Prediction of protein thermostability, ie. $\Delta\Delta G$ of folding, across three experimental datasets of diverse proteins, reported with Spearman correlation.

Widatalla et al., 2023), which are trained with similar MLM or AR objectives, but with no structural conditioning. We hypothesize that inverse folding models, using backbone geometry, gain a limited, but nonetheless effective, inductive bias for predicting binding affinity and stability changes upon mutation. From this, we aimed to evaluate if the addition of full-atomic context, and explicit modeling of the crucial sidechain interactions which often determine stability and binding affinity, can improve performance.

We evaluate FAMPNN (0.3Å) on three stability datasets,

five antibody-antigen binding affinity datasets, and two versions of a general protein-protein binding affinity dataset, against a range of state-of-the-art unsupervised and supervised methods for binding and stability prediction. We score the fitness of a mutation by masking the mutated position and sidechain, and taking the conditional likelihood of the mutant residue normalized by the conditional likelihood of the wild-type residue at the mutant position. Here, the advantage of our method is the ability to score a mutation conditioned not only on neighboring backbone and sequence context, but the full-atom structure of surrounding residues.

To evaluate zero-shot performance for prediction of protein-protein binding affinity, we evaluate on SKEMPIv2, a widely used curation of experimentally measured binding affinities for thousands of sequence variants across hundreds of protein-protein interactions totaling ∼7000 data points. Shown in Figure **5a**, FAMPNN significantly outperforms unsupervised models on the entire SKEMPIv2 dataset and, notably, even supervised models on the test split from Bushuiev et al. (Bushuiev et al., 2024). To evaluate zero-shot prediction of protein stability, we evaluate on S669, Megascale, and FireProtDB, which are datasets of experimentally measured changes in protein stability ($\Delta\Delta G$) to diverse native proteins and are widely-used benchmarks for stability predictors. Shown in Figure **5b**, FAMPNN modestly surpasses both ProteinMPNN and ESM-IF, consistently across all three stability datasets. Once again, shown in Figure **5c**, FAMPNN consistently outperforms state-of-the-art unsupervised methods ProteinMPNN and ESM-IF1 on the therapeutically relevant task of antibody-antigen binding affinity prediction. These results demonstrate the utility of FAMPNN for the stabilization of proteins and enhancement of protein-protein interaction.

### 5.4. Sidechain context improves performance

To evaluate whether full-atomic modeling improves sequence design performance, we ablated the two sources of full-atom information during model training: (1) full-atom conditioning, and (2) sidechain packing. We report performance of our implementation of MPNN and MPNN-GVP, with additions of (1) and/or (2) to MPNN-GVP (FAMPNN is the addition of both [1,2]). We report mean sequence recovery on the CATH 4.2 test set, training with 0Å and 0.3Å noise. Shown in Figure **6a**, we see incorporation of the sidechain packing objective improves sequence accuracy. Second, we see full-atomic conditioning also increases sequence accuracy, even more so when trained with 0.3Å noise. Both FAMPNN and the baseline model improve with additional sequence context, but interestingly, FAMPNN shows markedly stronger scaling (6% and 15% improvements at 90% context versus 2% and 8% at 0% context) from access to sidechain structural information unavailable to backbone-only models.

We also benchmark FAMPNN's ability to leverage both partial sequence and sidechain context to improve both sequence design and sidechain packing performance (Figure **6b, c**). In Figure **6b**, we find that along protein-protein interfaces, where modeling sidechain interactions are more critical, providing partial sidechain context with sequence improves accuracy over providing partial sequence context alone. Moreover, FAMPNN can more effectively leverage sidechain context as compared to LigandMPNN, which can also optionally condition on sidechain atoms as context. Furthermore, we show that FAMPNN can pack sidechains while conditioning on varying amounts of partial sequence or sidechain conformation context, with increasing context leading to better packing accuracy (Figure **6c**).

Lastly, we analyze the effect of full-atom conditioning on prediction of protein fitness. Shown in Figure **7**, we compare the average performance on fitness categories from Figure **5** of FAMPNN and a version of the model trained without full-atom conditioning and sidechain packing, but scaled to the same number of parameters. Here, we see improvements in fitness evaluation upon incorporation of full-atomic context in model training, specifically for the SKEMPIv2 dataset. Surprisingly, this does not translate to antibody-antigen binding prediction. This is likely due to the use of inaccurate input structures in this benchmark which potentially introduce adversarial structural context. (see Appendix **G.2.5**) Together, these results demonstrate that full-atom conditioning can serve as salient information for improved prediction of protein fitness.

# 6. Discussion

FAMPNN is a full-atom protein sequence design method, which models the per-token distribution of a residue's discrete amino acid identity and its continuous sidechain conformation. This distribution is learned with a combined categorical cross-entropy and diffusion loss, and we demonstrate that learning to model 3D sidechain coordinates improves sequence design performance. We outline an iterative process to jointly sample protein sequence and sidechain conformation, demonstrating its utility in designing sequences with higher self-consistency than in a single pass. We achieve state-of-the-art accuracy in sidechain packing RMSD, along with an accurate estimate of its own error, pSCE. Finally, we find that incorporation of full-atomic context improves sequence design performance, notably more so as the amount of structural context increases. This result underlies the significant improvement of FAMPNN over backbone-only models in protein fitness prediction, where we are often evaluating only single residue mutations, utilizing the rest of the full-atom structure as salient context.

While FAMPNN demonstrated the utility of sidechain conditioning and conformation modeling, adding the capacity

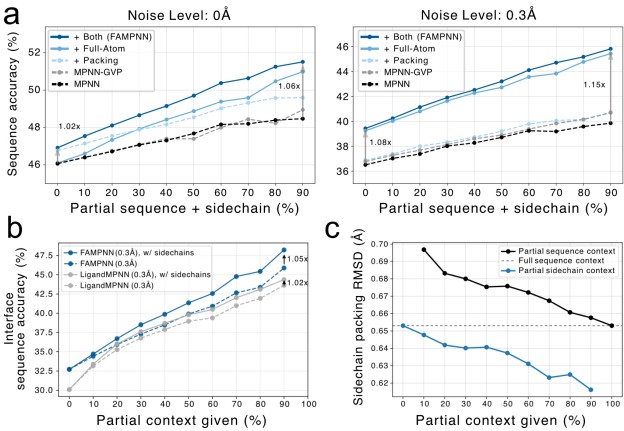

*Figure 6.* **(a)** Mean sequence accuracy on CATH 4.2 test set for various ablations of model training, conditioning and architecture with 0Å (left) and 0.3Å (right) of structural noise added during training. **(b)** Sequence accuracy at protein-protein interfaces vs. provided partial sequence context for FAMPNN and LigandMPNN. Solid lines use both sequence and sidechain context; dashed lines use sequence context only. **(c)** Average sidechain packing RMSD on CASP targets vs. partial context. Black: packing sidechains for known partial sequence context. Blue: packing remaining sidechains given partial sidechain context.

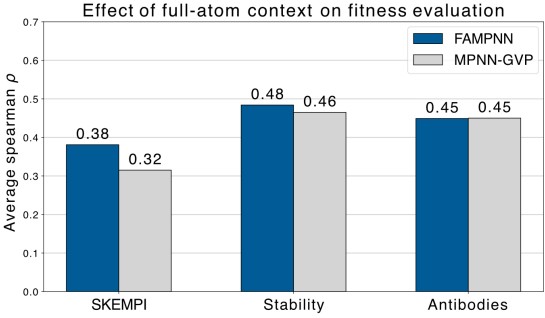

*Figure 7.* Ablation of full-atomic context on fitness evaluation performance. We report performance as spearman $\rho$ correlation with experimental measurements averaged across 10 checkpoints from 100-190k steps, for each category. We evaluate with the same FAMPNN (0.3Å) model, but with and without sidechain the coordinates of non-mutated residues

to condition on ligands (Dauparas et al., 2023) and model their conformation (Anishchenko et al., 2024), would expand the biological applications of this work to the design of sidechain-ligand interaction networks present in many functional proteins and enzymes. Second, following previous work that has demonstrated success in aligning inverse folding models to experimental data (Widatalla et al., 2024), FAMPNN can be similarly aligned to desired properties to further enrich design performance and fitness prediction. Lastly, because there are biological applications where a starting backbone structure is not available, future work will

investigate how FAMPNN can be modified to concurrently design protein backbone as well as sequence and sidechain conformation.

## Software

Code for FAMPNN is available at https://github.com/richardshuai/fampnn.

## Acknowledgements

We thank Tianyu Lu, Garyk Brixi, and Alex Chu for insightful discussions early in the project. We also thank Jin Sub Lee for providing the CASP datasets used for evaluation in (Lee et al., 2023). T.W acknowledges funding by the Stanford Graduate Fellowship. T.W and R.W.S acknowledge funding support from the NSF Graduate Research Fellowship (DGE-2146755). B.L.H. acknowledges funding support from Arc Institute. This research was funded in part by the Advanced Research Projects Agency for Health (ARPA-H) Agreement No. 1AY2AX000054 and D24AC00412-00. Additional support to P.-S.H are from Merck Research Laboratories (MRL) Scientific Engagement and Emerging Discovery Science (SEEDS) Program, Stanford Medicine Catalyst, and NIH (R01GM147893). The views and conclusions contained in this document are those of the authors and should not be interpreted as representing the official policies, either expressed or implied, of the U.S. Government. We are grateful to Justine Yuan for help with illustrations and visual presentation.

## Competing Interests

B.L.H. acknowledges outside interest in Prox Biosciences as a scientific cofounder. All other authors declare no competing interests.

## Impact Statement

This work has direct implications for discovery and optimization of therapeutics, specifically antibodies engineered to have high binding affinity to drug targets. Furthermore, full-atom sequence design can be particularly useful for the design of enzymes with improved activity and stability, with potential beneficial environmental use cases such as plastic degradation (Tournier et al., 2023) and heavy metal sequestration (Luo et al., 2024). However, similar to other advancements, there are risks that this technology can be used to harm human health, specifically via production of hazardous biomolecules. Hence, we strongly support adherence to ethical and biosafety guidelines for machine learning in biology as those set forth by Responsible AI X Biodesign.

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

# A. Benchmarks

## A.1. Fitness Evaluation Benchmarks

*Table 3.* Prediction of protein-protein binding affinity and stability. Bold values indicate the best results. Underlined values indicate second best.

| Dataset | Method | Spearman Correlation |
|---|---|---|
| SKEMPI | ProteinMPNN | 0.27 |
| | ESM-IF1 | 0.29 |
| | FAMPNN | **0.39** |
| SKEMPI-PPI | ProteinMPNN | 0.41 |
| | ESM-IF1 | 0.18 |
| | FAMPNN | **0.58** |
| | PPIFormer | 0.44 |
| | GEMME | 0.30 |
| | RDE Net | 0.24 |
| | MSA Transformer | 0.32 |
| Megascale | ProteinMPNN | 0.53 |
| | ESM-IF1 | 0.54 |
| | FAMPNN | **0.55** |
| FireProt | ProteinMPNN | 0.50 |
| | ESM-IF1 | 0.47 |
| | FAMPNN | **0.55** |
| S669 | ProteinMPNN | 0.30 |
| | ESM-IF1 | 0.31 |
| | FAMPNN | **0.41** |
| CR9114-H3 | ProteinMPNN | 0.33 |
| | ESM-IF1 | 0.37 |
| | FAMPNN | **0.42** |
| CR9114-H1 | ProteinMPNN | 0.44 |
| | ESM-IF1 | 0.22 |
| | FAMPNN | **0.53** |
| CR6261-H9 | ProteinMPNN | 0.47 |
| | ESM-IF1 | 0.45 |
| | FAMPNN | **0.55** |
| CR6261-H1 | ProteinMPNN | 0.46 |
| | ESM-IF1 | 0.43 |
| | FAMPNN | **0.56** |
| G6 | ProteinMPNN | 0.13 |
| | ESM-IF1 | 0.28 |
| | FAMPNN | **0.29** |

*Table 4.* Comparison with supervised baselines for stability prediction.

| Dataset | Method | Spearman Correlation | Pearson Correlation |
|---|---|---|---|
| Megascale (Test Set) | *Unsupervised* | | |
| | FAMPNN | 0.53 | 0.50 |
| | *Supervised* | | |
| | ThermoMPNN | **0.73** | **0.75** |
| FireProt | *Unsupervised* | | |
| | FAMPNN | 0.55 | 0.53 |
| | *Supervised* | | |
| | ThermoMPNN | **0.66** | **0.65** |
| | ProteinDPO | 0.6 | 0.6 |
| S669 | *Unsupervised* | | |
| | FAMPNN | 0.41 | 0.39 |
| | *Supervised* | | |
| | MSAesm | – | **0.54** |
| | MAESTRO | – | 0.50 |
| | PROSTATA | – | 0.49 |
| | ProteinDPO | – | 0.47 |
| | ACDC-NN | – | 0.46 |
| | ThermoMPNN | – | 0.43 |
| | DDGun3D | – | 0.43 |
| | INPS3D | – | 0.43 |
| | INPS-Seq | – | 0.43 |
| | ACDC-NN-Seq | – | 0.42 |
| | DDGun | – | 0.41 |
| | PremPS | – | 0.41 |
| | Dynamut | – | 0.41 |
| | SDM | – | 0.41 |
| | PoPMuSiC | – | 0.41 |
| | DUET | – | 0.41 |
| | RaSP | – | 0.39 |
| | ThermoNet | – | 0.39 |
| | I-Mutant3.0 | – | 0.36 |
| | Dynamut2 | – | 0.34 |

# B. Model training

## B.1. Structural noise

Previous work (Dauparas et al., 2022) has shown that the addition of random Gaussian noise to protein structure examples during training improves the self-consistency of model designs. In line with this, we trained two versions of FAMPNN, trained with 0 and 0.3Å noise, respectively. We add independent noise to each $x$, $y$, and $z$ coordinate by sampling from a normal distribution scaled by a factor of 0.3. Specifically, for each coordinate, the perturbation is drawn from $\mathcal{N}(0, \sigma^2)$, with standard deviation $\sigma = 0.3$.

## B.2. Training details

The CATH trained FAMPNN models were trained on a single NVIDIA H100 GPU with 80GB of RAM, with a batch size of 64 and fixed example size of 256 residues. The models were trained until 100k steps which required approximately 6-8 hours. The PDB trained models were trained on 4 NVIDIA H100 GPUs with 80GB of RAM per GPU, with a batch size of 8 per GPU and fixed example size of 1024 residues. To increase the effective batch size, 4 gradient accumulation steps

were taken for each backpropagation step, bringing the effective batch size to 128 examples. The models were trained until 300k steps which required approximately 3 days. For the PDB-trained models, we used the post-hoc EMA procedure from Karras et al. to choose EMA lengths after training was completed (Karras et al., 2024). Based on a combination of sequence accuracy and sidechain packing RMSD on the validation set, we chose an EMA length of 1% for the 0.3Å noise model and 25% for the 0.0Å noise model.

## B.3. Datasets

### B.3.1. CATH

We utilized the CATH 4.2 (Knudsen & Wiuf, 2010) S40 dataset which is a curation of domains extracted from the PDB with redundant domains (those with >40% homology) removed, with training, validation and test splits identical to Ingraham et al. (Ingraham et al., 2019). We set all examples to a fixed length by either cropping examples larger than the given fixed size, or padding examples which are smaller than the given fixed size. Cropping is done during training time by randomly sampling a single continuous span of the protein sequence. The start position of this crop is selected uniformly at random between the start of the sequence and the fixed size subtracted from the total sequence.

### B.3.2. PDB

We used publicly available (Wang et al., 2022) reproduced training splits curated from the entire Protein Data Bank (Berman et al., 2000) used to train AlphaFold3 (Abramson et al., 2024). This dataset includes a cutoff date of 2021-09-30, such that no structures released after this date are included. This dataset was clustered on the chain level at 40% sequence homology for proteins. Then, interface-based clustering is performed as a join on the cluster IDs of the corresponding chains in the interface, such that interfaces $i$ and $j$ are in the same interface cluster $C$ interface only if their constituent chain pairs $\{I_1, I_2\}, \{J_1, J_2\}$ have the same chain cluster pairs $\{C_1^{\text{chain}}, C_2^{\text{chain}}\}$. Additional details regarding curation of this dataset have been previously described (Abramson et al., 2024). To (1) prioritize training of the PDB model to learning to design multichain proteins and (2) ensure protein structure was provided with correct context, we removed every single-chain example which is present as a chain within a multichain example. Second, to maximize the amount of training data whilst preventing redundant training examples, we randomly sample a single example from each training cluster at each epoch. In line with training of AlphaFold-Multimer, we supplement our contiguous cropping strategy on the CATH dataset, with a previously described (Evans et al., 2022) interface spatial cropping strategy to maximize interface coverage for multichain examples. Our strategy, described in Algorithm **1**, is slightly simpler, given that our multichain examples contain only two chains. Our spatial cropping strategy is identical to Alphafold-Multimer. In line with (Evans et al., 2022) we randomly sample with equal probability whether to apply spatial cropping or contiguous cropping to a given multichain example.

---

**Algorithm 1** Multimer Contiguous Crop

**Notation:** $l_1, l_2$: chain lengths; $k$: desired crop size; $m$: binary mask; $n$: total length; $c_i$: crop size for chain $i$; $s_i$: start position for chain $i$

1: **function** MULTIMERCROP($l_1, l_2, k$)
2:   $n \leftarrow l_1 + l_2$
3:   $m \leftarrow [0]^n$                       ▷ Initialize mask
4:   $c_1 \leftarrow \text{uniform}(\min(l_1, \max(0, k - l_2)), \ \min(l_1, k) + 1)$
5:   $c_2 \leftarrow k - c_1$
6:   $s_1 \leftarrow \text{uniform}(0, \ n - k)$
7:   $m[s_1 : s_1 + c_1] \leftarrow 1$
8:   **if** $c_2 > 0$ **then**
9:     $s_2 \leftarrow \text{uniform}(s_1 + c_1, \ n - c_2)$
10:    $m[s_2 : s_2 + c_2] \leftarrow 1$
11:   **end if**
12:   **return** $m$                ▷ Binary mask $m$ for selected residues
13: **end function**

---

## C. Masked language modeling scheme

Because our method generates both sequence and sidechain coordinates in a per-token manner, we can view the full-atom sequence design procedure through a generative masked language modeling framework. Each token is both a residue's sequence identity and its sidechain conformation. Similar to MaskGIT and ESM3 (Chang et al., 2022; Hayes et al., 2024), we train with randomly selected masking rates. Here, we draw from a concave schedule:

$$t = \sqrt{u}, \quad u \sim \text{Uniform}(0, 1)$$

where each residue in the sequence is kept with probability $t$. This prioritizes mostly unmasked tokens during training, focusing learning on filling in missing tokens given surrounding context.

### C.1. Per-token modeling of sequence and sidechains

Here, we describe how we model a residue's sequence and its sidechain conformation as a single token. Following MaskGIT's notation, let $\mathbf{Y} = [y_i]_{i=1}^N$ be a sequence of $N$ tokens, with a corresponding binary mask $\mathbf{M} = [m_i]_{i=1}^N$, where $m_i = 1$ if $y_i$ is masked. Define the **masked set** $M = \{i \mid m_i = 1\}$ and its complement $\overline{M}$ of unmasked indices. The model learns to predict masked tokens conditioned on unmasked tokens:

$$p_\theta(\mathbf{Y}_M \mid \mathbf{Y}_{\overline{M}}).$$

For categorical tokens, this corresponds to a standard masked language modeling (MLM) objective:

$$\mathcal{L}_{\text{MLM}} = \mathbb{E}_{\mathbf{Y}, M} \sum_{i \in M} \log p_\theta(y_i \mid \mathbf{Y}_{\overline{M}}).$$

In our problem, each token $y_i$ consists of both a sequence identity $s_i$ (categorical) and a sidechain conformation $x_i$ (continuous). We therefore decompose the token distribution as:

$$p(y_i \mid \mathbf{Y}_{\overline{M}}) = p(s_i, x_i \mid \mathbf{Y}_{\overline{M}}) = p(s_i \mid \mathbf{Y}_{\overline{M}})p(x_i \mid s_i, \mathbf{Y}_{\overline{M}})$$

This factorization requires learning both:

1. Sequence prediction: $p(s_i \mid \mathbf{Y}_{\overline{M}}) \rightarrow$ trained with the **MLM loss** $\mathcal{L}_{\text{MLM}}$ (Devlin, 2018).

2. Sidechain prediction: $p(x_i \mid s_i, \mathbf{Y}_{\overline{M}}) \rightarrow$ trained with the **diffusion loss** $\mathcal{L}_{\text{diff}}$ from Li et al. (Li et al., 2024), while conditioning on $s_i$.

To learn both objectives simultaneously, we simply sum the losses, yielding our total loss for the main model (excluding the confidence module):

$$\mathcal{L}_{\text{total}} = \mathcal{L}_{\text{MLM}} + \mathcal{L}_{\text{diff}}$$

We did not experiment with relative weightings of the losses on each objective.

## D. FAMPNN architecture

### D.1. Full-atom representation

In FAMPNN we represent residues in a combination of the similar `atom37` and `atom14` formats.

In the `atom37` format, each residue is represented as a fixed size matrix of size $37 \times 3$. Each row corresponds to the 3D coordinates of all 37 possible atom types. 4 of these rows correspond to the backbone atoms N, C$\alpha$, C, and O, while the remaining 33 rows correspond to the residue's sidechain atoms. For sidechains where a particular atom type is not present, the row for that atom type is a "ghost atom" and is set to the residue's C$\alpha$ position.

The `atom14` format is very similar to `atom37` but is condensed to only the maximum number of atom types a single amino-acid can possess. Thus, other than the backbone atoms N, C$\alpha$, C, and O, which are shared between all amino-acids,

---

**Algorithm 2** Full-atom Encoder

---

**Notation:** $X$: full-atom coordinates; $S$: sequence; $M$: sequence mask; $R$: residue indices; $C$: chain indices; $h$: hidden dimension; $k$: number of neighbors, $E_{idx}$: indices of connected nodes, $L_{inv}, L_{eq}$: number of invariant and equivariant layers respectively, $V_{node}$: vector node features, $V_{edge}$: vector edge features, $s_{node}$: scalar node features, $s_{edge}$: scalar edge features.

1: **function** FULLATOMENCODER($X, S, M, R, C, h, k, L_{inv}, L_{eq}$) $\qquad\qquad\triangleright X \in \mathbb{R}^{n \times 14 \times 3}, S \in \mathbb{N}^n, M \in \{0,1\}^n$

        # Backbone structure featurization
2:      $s_{edge}, E_{idx} \leftarrow$ InvariantBackboneFeatures($X_{bb}, R, C$)
3:      $h_V \leftarrow \mathbf{0}^{n \times h}$
4:      $h_E \leftarrow$ Linear($s_{edge}$)

        # Invariant backbone encoder layers
5:      **for** $l = 1$ to $L_{inv}$ **do**
6:          $h_V, h_E \leftarrow$ MPNNEncoder($h_V, h_E, E_{idx}, M$)
7:      **end for**

        # Full-atom structure and sequence featurization
8:      $h_E \leftarrow$ Concat[$h_E$, Linear(InvariantFullAtomFeatures($X, R, C, E_{idx}$)]
9:      $h_S \leftarrow$ EmbedSeq($S$)
10:     $h_{ES} \leftarrow$ ConcatNeighbors($h_S, h_E, E_{idx}$)
11:     $h_{ESV} \leftarrow$ ConcatNeighbors($h_V, [h_{ES}; h_{E2}], E_{idx}$)

        # Invariant full-atom encoder layers
12:     **for** $l = 1$ to $L_{inv}$ **do**
13:         $h_V, h_{ESV} \leftarrow$ MPNNDecoder($h_V, h_{ESV}, M, E_{idx}$)
14:     **end for**

        # Equivariant full-atom featurization
15:     $V_{node} \leftarrow$ GetUnitVectors($X$)
16:     $V_{edge} \leftarrow$ GetEdgeVectors($X, E_{idx}$)

        # Invariant full-atom featurization
17:     $s_{node} \leftarrow$ GetDihedrals($X$)
18:     $s_{edge} \leftarrow$ GetDistances($X, E_{idx}$)

        # Embed GVP features
19:     $h_V^{inv}, h_V^{eq} \leftarrow$ GVP($V_{node}, s_{node}$)
20:     $h_E^{inv}, h_E^{eq} \leftarrow$ GVP($V_{edge}, s_{edge}$)

        # Merge invariant features
21:     $h_V^{inv} \leftarrow$ Linear($h_V^{inv} + h_V$)
22:     $h_E^{inv} \leftarrow$ Linear($h_E^{inv} + h_ESV$)

        # Equivariant full-atom encoder layers
23:     **for** $l = 1$ to $L_{eq}$ **do**
24:         $h_V^{inv}, h_V^{eq}, h_E^{inv}, h_E^{eq} \leftarrow$ GVP($h_V^{inv}, h_V^{eq}, h_E^{inv}, h_E^{eq}$)
25:     **end for**

26:     $h_V \leftarrow$ Concat[$h_V^{eq} \cdot R^{-1}; h_V^{inv}$]
27:     $L \leftarrow$ Linear($h_V$)

28:     **return** $L, h_V$ $\qquad\qquad\qquad\qquad\qquad\triangleright$ Sequence logits $L \in \mathbb{R}^{n \times 20}$, node embeddings $h_V$
29: **end function**

---

the atom type of the $i$th position in one residue's atom14 representation may not be the same atom type as another residue. Specifically, a residue is represented as a fixed size matrix of size 14 × 3. Each row corresponds to the 3D coordinates of the maximum number of possible atom types a single residue can possess. 4 of these rows correspond to the backbone atoms N, C$\alpha$, C, and O, while the remaining 10 rows correspond to the residue's sidechain atoms. As with atom37, for sidechains where a particular atom type is not present, the row for that atom type is set to the residue's C$\alpha$ position.

We opt to use atom14 for full-atom conditioning in the sequence design module, and atom37 for Euclidean sidechain diffusion.

### D.2. MPNN-GVP full-atom encoder

We represent full-atom protein structure as a graph, encoded with a graph neural network (GNN) using a hybrid MPNN-GVP architecture (Dauparas et al., 2022; Jing et al., 2020). This architecture consists of three primary components, which are an invariant backbone encoder, an invariant full-atom encoder, and an equivariant full-atom encoder, with respect to any arbitrary composition of global rotations and reflections in protein coordinates.

The first two components build off the architecture of ProteinMPNN (Dauparas et al., 2022). ProteinMPNN featurizes backbone structure as a k-NN graph, with a set of nodes $\mathcal{V}$ that represent each protein residue, and edges $\mathcal{E} = \{e_{i \to j}\}$ which are defined for the 48 nearest neighbors of each residue by C$\alpha$ distance. All node representations $\mathcal{V}$ are initialized with 0s, and edges in $\mathcal{E} = \{e_{i \to j}\}$ are initialized with an encoding of distances between the C, C$\alpha$, N, O and C$\beta$, atoms of residues $i$, $j$ in terms of Gaussian radial basis functions (RBF). ProteinMPNN includes an initial "structure encoder" trunk consisting of three previously described (Ingraham et al., 2019) invariant MPNN layers, which update both edge and node representations MPNNEncoder$(v_i, e_{ij}) = (v_i', e_{ij}')$. This output is passed to a sequence decoder comprised of three additional invariant MPNN layers. In addition to the output of the structure encoder, causally-masked one-hot encodings of sequence identity are concatenated to the edge representation $\mathcal{E}'$. The sequence decoder MPNNDecoder$(v_i', [e_{ij}', s_i]) = (v_i'')$, then updates only node representations, which are passed into a final output head for sequence prediction.

Our initial component, the invariant backbone encoder, is identical to MPNNEncoder, encoding the backbone structure only. However, for our second component, the invariant full-atom encoder, we replace MPNNDecoder with a full-atom encoder, which is identical to the backbone encoder, but with expanded featurization to all atoms. To do this, for all $\mathcal{E} = \{e_{i \to j}\}$, we concatenate an RBF encodings of the distances between the backbone atoms (C, C$\alpha$, N, O) of residue $i$ and *all* neighboring atoms in residue $j$. Here, we compress the composition of neighboring atoms to the atom14 representation. Similar to MPNNDecoder, we concatenate one-hot encodings of sequence identity, but remove the causal mask, as FAMPNN is trained with an MLM objective, and sequence positions are randomly masked with a corresponding token.

The third and final component of the model, the equivariant full-atom encoder, consists of modified Geometric Vector Perceptrons (GVP) from Jing et al . GVP layers consist of an equivariant track for learning vector features and an invariant track for learning scalar features. Concretely, given a tuple $(\mathbf{s}, \mathbf{V})$ of scalar features $\mathbf{s} \in \mathbb{R}^n$ and vector features $\mathbf{V} \in \mathbb{R}^{\nu \times 3}$, GVP computes new features $(\mathbf{s}', \mathbf{V}') \in \mathbb{R}^m \times \mathbb{R}^{\mu \times 3}$. In FAMPNN, in addition to the forward and reverse unit vectors in the directions of $C_{\alpha_{i+1}} - C_{\alpha_i}$ and $C_{\alpha_{i-1}} - C_{\alpha_i}$ used in Jing et al. as vector-valued node features, we use the equivariant track to encode unit vectors from $C_{\alpha_i}$ to all other atoms in residue $i$. For the equivariant edge features $e_{ij}$, in addition to a unit vector in the direction $\mathbf{C}_{\alpha_j} - \mathbf{C}_{\alpha_i}$, we include unit vectors from $\mathbf{C}_{\alpha_i}$ to all atoms in residue $j$, again, in the compressed atom14 representation. The invariant track embeds the sine and cosine of the backbone dihedral angles of residue $i$ as node features, and sinusoidal encodings of the value $j - i$, as in Vaswani et al. (Vaswani et al., 2023), and RBFs of the distance $\|C_{\alpha_j} - C_{\alpha_i}\|_2$ as edge features, as previously described (Jing et al., 2020). We additionally incorporate distances from $\mathbf{C}_{\alpha_i}$ to all atoms in residue $j$.

We initialize the invariant-track of GVP with node and edge representation output from the invariant full-atom encoder. Given the sequence identity of a protein is invariant to global rotations and reflections, the final GVP layers return an invariant output by multiplying all vector features with the inverse rotation matrix $R \in \mathbb{R}^{3 \times 3}$ which defines the frame of the input structure. We then concatenate these outputs with the invariant scalar hidden state. This representation is then passed to both a final output head for sequence prediction, and the sidechain diffusion module for invariant Euclidean denoising of sidechain atoms.

## D.3. Sidechain diffusion

### D.3.1. INVARIANT SIDECHAIN REPRESENTATION

During sidechain diffusion, we transformed the sidechain of each residue into a local reference frame defined by its backbone atoms. To construct frames, we used the OpenFold implementation of Algorithm 21 from AlphaFold2, which defines a frame from 3 points (Ahdritz et al., 2024; Jumper et al., 2021). The pseudocode is reproduced here in Algorithm 3 for the reader's reference. To construct a frame for each backbone, we use N as $\vec{x}_1$, C$\alpha$ as $\vec{x}_2$, and C as $\vec{x}_3$. Transforming each sidechain into its local backbone frame allows the sidechain diffusion process for each residue to be invariant to global rotations and translations of the input backbone.

---

**Algorithm 3** Rigid from 3 points using the Gram-Schmidt process

---

1: **function** RIGIDFROM3POINTS($\vec{x}_1$, $\vec{x}_2$, $\vec{x}_3$)
2:     $\vec{v}_1 \leftarrow \vec{x}_3 - \vec{x}_2$
3:     $\vec{v}_2 \leftarrow \vec{x}_1 - \vec{x}_2$
4:     $\vec{e}_1 \leftarrow \vec{v}_1 \, / \, \|\vec{v}_1\|$
5:     $\vec{u}_2 \leftarrow \vec{v}_2 - \vec{e}_1 \, (\vec{e}_1^\top \vec{v}_2)$
6:     $\vec{e}_2 \leftarrow \vec{u}_2 \, / \, \|\vec{u}_2\|$
7:     $\vec{e}_3 \leftarrow \vec{e}_1 \times \vec{e}_2$
8:     $R \leftarrow \text{concat}(\vec{e}_1, \vec{e}_2, \vec{e}_3)$
9:     $\vec{t} \leftarrow \vec{x}_2$
10:     **return** $(R, \vec{t})$
11: **end function**

---

### D.3.2. DIFFUSION SCHEME

As discussed in Section 4.3.1, our diffusion process uses the variance-exploding EDM scheme from Karras et al. (Karras et al., 2022). We computed $\sigma_{\text{data}} = 0.66$ by taking the standard deviation of sidechain coordinates in their local frame from a random batch of 1000 examples from the training dataset. An overview of the parameters we used can be found in Table D.3.2. For sampling, we run a full trajectory using 50 steps of diffusion with a step scale $\eta = 1.5$.

| Parameter | Value |
|---|---|
| $\sigma_{\text{min}}$ | 0.01 |
| $\sigma_{\text{max}}$ | 80 |
| $\sigma_{\text{data}}$ | 0.66 |
| $\rho$ | 7 |
| $P_{\text{mean}}$ | 1.5 |
| $P_{\text{std}}$ | 1.0 |

*Table 5.* Overview of EDM parameters used in this work.

---

**Algorithm 5** Conditioned MLP Block

---

**Notation:**   $x$: input tensor; $c$: per-token conditioning;
1: **function** CONDITIONEDMLPBLOCK($x$, $c$)
      # Pointwise feedforward with AdaLN conditioning
2:     $(\beta, \gamma, \alpha) \leftarrow \text{Linear}\big(\text{SiLU}(c)\big)$          $\triangleright$ Split into shift $\beta$, scale $\gamma$, gate $\alpha$
3:     $x_{\text{skip}} \leftarrow \text{LayerNorm}(x)$          $\triangleright$ No learnable bias or scaling
4:     $x_{\text{skip}} \leftarrow x_{\text{skip}} \times (1 + \gamma) + \beta$
5:     $x_{\text{skip}} \leftarrow \text{Dropout}\Big(\text{Linear}\big(\text{LayerNorm}\big(\text{Dropout}\big(\text{GELU}\big(\text{Linear}(x_{\text{skip}})\big)\big)\big)\big)\Big)$     $\triangleright$ MLP block
6:     $x \leftarrow x + \alpha \times x_{\text{skip}}$
7:     **return** $x$
8: **end function**

---

---

**Algorithm 4** Sidechain diffusion MLP

---

**Notation:** $X_{\text{scn}}^{\text{noisy}}$: noisy local sidechain coordinates; $v_i$: node embeddings; $s_i$: predicted one-hot encoded sequence; $\sigma$: noise level;

1: **function** SIDECHAINMLP($X_{\text{scn}}^{\text{noisy}}, v_i, s_i, \sigma$)
2:     $x_i \leftarrow \text{reshape}(X_{\text{scn}}^{\text{noisy}})$                                                   ▷ Flatten atom and xyz dimension
      # Embed inputs
3:     $x_i \leftarrow \text{concat}(x_i, s_i)$
4:     $x_i \leftarrow \text{Linear}(x_i)$
      # Embed conditioning inputs
5:     $\sigma_{\text{embed}} \leftarrow \text{NoiseEmbedder}(\sigma)$
6:     $v_{\text{embed}} \leftarrow \text{Linear}(v_i)$
7:     $c_i \leftarrow \sigma_{\text{embed}} + v_{\text{embed}}$
      # Run MLP blocks
8:     **for** $\ell = 1$ to $4$ **do**
9:         $x_i \leftarrow \text{ConditionedMLPBlock}(x_i, c_i)$                    ▷ Apply AdaLN + pointwise feedforward
10:    **end for**
      # Project back into coordinate space
11:    $x_i \leftarrow \text{FinalLayer}(x_i, c)$                                ▷ Unflatten atom and xyz dimension
12:    $X_{\text{scn}}^{\text{pred}} \leftarrow \text{reshape}(x_i)$
13:    **return** $X_{\text{scn}}^{\text{pred}}$
14: **end function**

---

**Algorithm 6** Final Layer

---

**Notation:** $x$: input tensor; $c$: conditioning tensor;

1: **function** FINALLAYER($x, c$)
      # Final projection with AdaLN conditioning
2:     $(\beta, \gamma) \leftarrow \text{Linear}(\text{SiLU}(c))$                          ▷ Split into shift $\beta$, scale $\gamma$
3:     $x \leftarrow \text{LayerNorm}(x)$                                 ▷ No learnable bias or scaling
4:     $x \leftarrow x \times (1 + \gamma) + \beta$
5:     $x \leftarrow \text{Linear}(x)$                                  ▷ Project to final output
6:     **return** $x$
7: **end function**

---

### D.3.3. SIDECHAIN DIFFUSION MLP

We parametrize the denoiser $D_\theta$ with a multilayer perceptron (MLP) that uses adaptive layer normalization (AdaLN) for conditioning, following Li et al. (Li et al., 2024; Peebles & Xie, 2023). As conditioning input for residue $i$, we use node embedding $i$, the current noise level, and the predicted amino acid type for residue $i$ (Algorithm 4). To embed noise levels, we use the implementation of the timestep embedder from Peebles et al., which uses a 256-dimensional frequency embedding followed by a 2-layer MLP (Algorithm 4, Line 5).

Crucially, each of the conditioning inputs to the sidechain diffusion MLP is invariant to global rotations and translations of the backbone. Furthermore, the sidechain coordinates fed into the MLP are represented in a local reference frame defined by the backbone atoms. Therefore, any global SE(3) transformation of the backbone leaves all inputs to the MLP unchanged. As a result, each denoising step made with $D_\theta$ is invariant to global rotations and translations of the input backbone.

### D.4. Confidence module

As discussed in Section 4.3.2, at train time, we trained a confidence module to predict per-atom sidechain errors given a packed structure. This module is trained to take in an input sequence, predicted sidechains, and node and edge embeddings from the full-atom encoder to predict a per-atom error for all sidechains (Algorithm 7). Similar to AlphaFold3 (Abramson et al., 2024), during training, we run a diffusion rollout to generate sidechain coordinates, and we apply a stop gradient to all confidence module inputs so that the confidence loss does not affect the training of the main model. For computational

efficiency, we found it sufficient to train the confidence module roughly once every 8 training steps: in practice, at each training iteration, we sample $u \sim \text{Uniform}(0,1)$ and only train the confidence module if $u \leq 1/8$.

---

**Algorithm 7** Sidechain Confidence Prediction

**Notation:** $X_{\text{scn}}$: predicted local sidechain coordinates; $v_i$: node embeddings; $e_{ij}$: edge embeddings; $s_i$: predicted sequence;

1: **function** PREDICTPSCE($X_{\text{scn}}, v_i, e_{ij}, s_i$)
     # Embed full-atom encoder outputs
2:     **for** $\ell = 1$ to 3 **do**
3:        $v_i, e_{ij} \leftarrow \text{MPNNDecoder}(v_i, [e_{ij}, s_i])$
4:     **end for**
     # Embed aatype and sidechain coordinates
5:     $v_i \leftarrow v_i + \text{Linear}(s_i)$
6:     $v_i \leftarrow v_i + \text{Linear}(X_{\text{scn}})$
     # Predict per-atom confidence
7:     $p_l \leftarrow \text{Softmax}\big(\text{Linear}(\text{SiLU}(\text{Linear}(v_i)))\big)$             ▷ Predict per-atom bin probabilities
8:     $b \leftarrow \text{Linspace}(0.0625, 4.0625, 33)$                        ▷ Get bin centers
9:     $e_l^{\text{pSCE}} \leftarrow p_l^\top b$                                ▷ Compute expected confidence per atom
10:    **return** $e_l^{\text{pSCE}}$
11: **end function**

---

### D.4.1. PREDICTED SIDECHAIN ERROR (PSCE)

During training, we binned the true per-atom sidechain error in 33 evenly spaced bins between 0Å to 4Å. The confidence module loss was then computed as a categorical cross entropy loss between the confidence model output and the binned errors. At inference, we obtain a single value for the predicted sidechain error by computing an expectation across all bins using the bin centers (Algorithm 7, Line 8 and 9).

### D.5. Sampling Speed

Regarding inference costs, when evaluating our CATH trained model on the CATH 4.2 test set, we achieve a single-step and five-step sampling speed of 0.03s and 0.11s per sample on a single H100 GPU.

## E. Self-consistency evaluation

### E.1. Dataset

We follow after ProteinBench (Ye et al., 2024b) and benchmark self-consistency on a set of RFdiffusion *de novo* backbones from lengths 100 to 500. For each length in $\{100, 200, 300, 400, 500\}$, we generated 100 samples from RFdiffusion using the default parameters. For structure prediction, we use single-sequence AlphaFold2 with 3 recycles as implemented in ColabDesign (Mirdita et al., 2022).

### E.2. Self-consistency benchmarking details

A full table of sequence design methods that we benchmarked on our *de novo* backbone dataset can be found at Appendix Table 6. Following ProteinBench, we use a sampling temperature of 0.1 for all methods unless otherwise stated. We compared the proposed method against several sequence design baselines, using the official recommended hyperparameters wherever possible:

- **FAMPNN (0.3Å, 0.0Å)**: We used the PDB-trained FAMPNN model to sample sequences using 100 iterative unmasking steps. To minimize the risk of exposure bias during iterative sampling of sidechains, at each step, we condition the model only on previously generated sidechains where the predicted sidechain error is less than 0.3Å.

- **ChromaDesign**: We used the publicly available weights from the official GitHub repository with the default sampling

*Table 6.* Median self-consistency achieved by sequence design methods on 100 *de novo* backbones generated by RFdiffusion for each length in {100, 200, 300, 400, 500}. Bolded values indicate the best results.

|  | Length 100 | | Length 200 | | Length 300 | | Length 400 | | Length 500 | |
|---|---|---|---|---|---|---|---|---|---|---|
|  | scTM (↑) | pLDDT (↑) | scTM (↑) | pLDDT (↑) | scTM (↑) | pLDDT (↑) | scTM (↑) | pLDDT (↑) | scTM (↑) | pLDDT (↑) |
| FAMPNN (0.3Å) | **0.968** | **93.00** | **0.967** | **91.27** | **0.938** | 83.45 | 0.760 | 74.73 | 0.545 | 61.73 |
| FAMPNN (0.0Å) | 0.896 | 88.99 | 0.890 | 81.87 | 0.703 | 67.86 | 0.602 | 63.80 | 0.471 | 55.49 |
| ProteinMPNN | 0.964 | 92.45 | 0.960 | 89.65 | 0.935 | **85.66** | **0.784** | **76.81** | **0.578** | **62.36** |
| ESM3 (1.4B) | 0.941 | 90.61 | 0.829 | 77.21 | 0.620 | 63.70 | 0.481 | 58.89 | 0.419 | 52.76 |
| ESM-IF1 | 0.605 | 85.57 | 0.678 | 68.20 | 0.468 | 60.04 | 0.493 | 58.21 | 0.348 | 52.71 |
| LM-Design | 0.662 | 82.48 | 0.487 | 64.55 | 0.407 | 57.08 | 0.419 | 53.72 | 0.356 | 52.62 |
| AttnPacker | 0.586 | 85.43 | 0.563 | 59.06 | 0.463 | 50.95 | 0.414 | 48.00 | 0.371 | 44.65 |

settings (t=0.5).

- **ProteinMPNN**: We used the 0.2Å checkpoint (`vanilla_model_weights/v_48_020.pt`), with all other sampling parameters at their defaults.

- **ESM3**: We used the publicly available ESM3-open model with 1.4B parameters, with number of unmasking steps equal to the length of the target backbone.

- **ESM-IF1**: We used the publicly available model weights and ran `sample_sequences.py` script from the official GitHub repository.

- **LM-Design**: We used the CATH-trained ESM-2 650M version of the model from the official GitHub repository (`lm_design_esm2_650m`).

- **AttnPacker**: We used the AttnPacker+Design model variant that can also condition on sidechain rotamers from the official GitHub repository. Following (McPartlon & Xu, 2023), we sampled sequences from the model using Gibbs sampling with 20 transitions per design, a 15% re-sampling rate, and a linear temperature decay from 0.5 to 0.1.

### E.3. Diversity

In addition to the self-consistency of designed sequences for the *de novo* generated backbones, we compared their diversity by reporting the average pairwise sequence similarity of the designs.

### E.4. Sampling Time

Finally, regarding inference costs, when evaluating our CATH model on the test-set, we achieve a single-step sampling speed of 0.03s and 0.11s per sample on a single H100 GPU.

*Table 7.* Average pairwise similarity of PDB trained model on 100 *de novo* backbones generated by RFdiffusion for each length in {100, 200, 300, 400, 500}.

|  | Length 100 Sim (↓) | Length 200 Sim (↓) | Length 300 Sim (↓) | Length 400 Sim (↓) | Length 500 Sim (↓) |
|---|---|---|---|---|---|
| FAMPNN (0.3Å) | 13.4% | **9.1**% | **8.4**% | **7.9**% | **7.8**% |
| FAMPNN (0Å) | 14.4% | 10.1% | 9.3% | 8.8% | 8.6% |
| ProtieinMPNN | 13.9% | 9.7% | 9.0% | 8.6% | 8.4% |
| ChromaDesign | **11.1**% | 9.2% | 8.7% | 8.4% | 8.3% |
| ESM-IF1 | 16.9% | 10.9% | 11.0% | 10.0% | 9.5 |
| ESM3 | 20.2% | 15.4% | 12.3% | 12.6% | 12.1% |

## F. Sidechain packing evaluation details

As mentioned in Section 5.2, following FlowPacker (Lee & Kim, 2024), we compared our sidechain packing performance to other methods on CASP13, 14, and 15 targets and used MMseqs2 `easy-search` to remove all CASP13-15 homologues

from our training and validation dataset with a cutoff of 40% similarity. The CASP13 and CASP14 test sets were obtained directly from the official AttnPacker GitHub repository (McPartlon & Xu, 2023), and the CASP15 targets were downloaded from the CASP data archive.

For computing the average sidechain packing RMSD on a given dataset, we first computed the per-residue RMSD for each residue in a protein. Then, we averaged these per-residue RMSDs over all residues in the protein to obtain a per-protein RMSD. Finally, we took the mean across all proteins in the dataset. For core and surface residue RMSDs (Table 2), we followed AttnPacker's definitions: core residues had at least 20 $C\beta$ atoms within 10Å, while surface residues had at most 15 $C\beta$ atoms within 10Å.

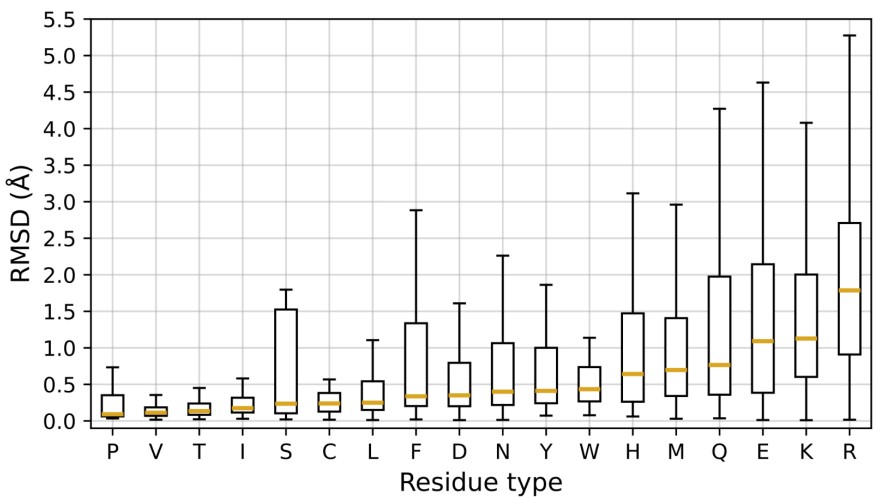

*Figure 8.* Sidechain packing RMSD distributions per residue type, computed from the CASP15 test set.

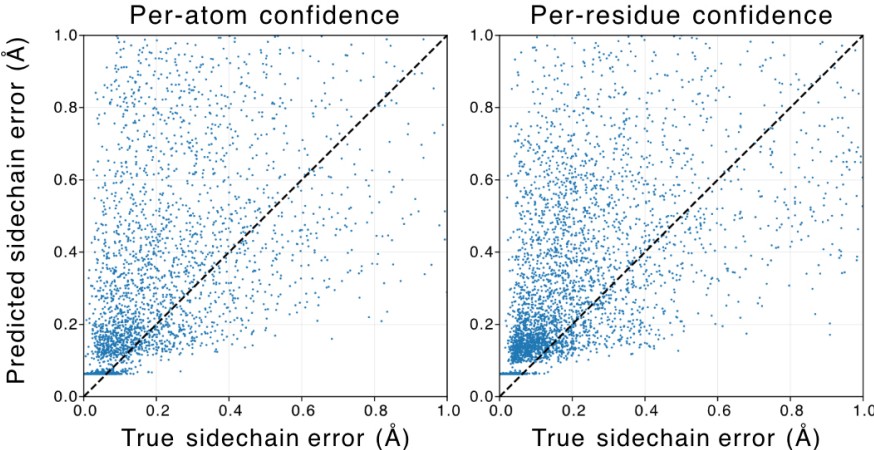

*Figure 9.* Predicted sidechain error compared with true sidechain errors, both per-atom (left) and per-residue (right), zoomed in to errors within 1Å.

# G. Fitness evaluation

## G.1. Model comparisons

### G.1.1. FAMPNN

We evaluate mutational fitness with FAMPNN by supplying the entire complex full-atom structure and sequence as input. However, we mask the sidechain and sequence identity of a given mutated residue. Because FAMPNN can additionally perform sidechain packing, when evaluating a single-mutant within a set of simultaneous mutations, we pack the sidechains of accompanying mutants. We then sum the log likelihood of each mutant, and subtract from the summed wild-type likelihoods.

### G.1.2. ESM-IF1

To run ESM-IF1, we followed instructions and code from the ESM-IF1 Github (Hsu et al., 2022), summing the log-likelihood of simultaneous mutations given the entire complex backbone-structure and sequence as input. However we took the additional step of normalizing with the summed wild-type likelihood of said mutated residues.

### G.1.3. PROTEINMPNN

For running ProteinMPNN we similarly provided the the entire complex backbone-structure and sequence as input, using the `score` function provided in the LigandMPNN Github to attain log-likelihoods (Dauparas et al., 2023). We then perform the same strategy as ESM-IF1 which is summation across mutated positions, and normalization with summed wild-type likelihood.

## G.2. Evaluation datasets

Unless specifically indicated, we report performance using an all vs. all calculation of the Spearman correlation between model predictions and dataset labels.

### G.2.1. MEGASCALE

We used a curated, de-duplicated, version of the entire Megascale dataset from Diekhaus et. al (Dieckhaus et al., 2024), combining the train, validation and test splits into a single dataset given only unsupervised methods were evaluated. This resulted in a final dataset size of 272,712 experimental $\Delta\Delta G$ data points across 298 unique proteins.

### G.2.2. SKEMPIv2

SKEMPIv2 is database of binding free energy changes upon single point mutations within a variety of protein complex interfaces (Jankauskaitė et al., 2018). Due to inability of either ProteinMPNN, ESM-IF1, or all three models (FAMPNN included) to handle some examples with non-canonical amino-acids, examples were removed for a final dataset size of 6,649 data points. We additionally evaluate on a test subset of SKEMPIv2 recently curated in Bushuiev et al to address significant data leakage issues in supervised models trained on previously proposed data splits (Bushuiev et al., 2024). We also report performance of supervised and MSA-based models on this dataset originally reported in Bushiev et al, with the exception of ProteinMPNN, which we independently evaluated, as this baseline was not included.

### G.2.3. FIREPROTDB

The FireProt database is a curated dataset of changes in free energy ($\Delta\Delta G$) for 3,438 single mutations for 100 unique proteins (Stourac et al., 2020). Due to inability of either ProteinMPNN, ESM-IF1, or all three models (FAMPNN included) to handle some examples with non-canonical amino-acids, 18 examples were removed for a total of 3420 examples.

### G.2.4. S669

The S669 dataset contains experimentally measured $\Delta\Delta G$ values of 669 single mutations of 94 proteins (Pancotti et al., 2022). Due to the presence of non-canonical amino acids, 4 variants were removed from this dataset.

G.2.5. ANTIBODY-ANTIGEN BINDING AFFINITY

The CR9114 (H1,H3), CR6261 (H1,H9) and G6 (VEGF-A) antibody-antigen binding affinity datasets were sourced from Shanker et al, from which details on dataset availability and the input structures used can be found (Shanker et al., 2024). The CR9114 dataset includes all possible combinations of 16 amino-acid substitutions, whereas CR6261 includes all possible combinations of 11 amino-acid substitutions, totaling 65,536 and 2,048 sequences respectively. Each of these libraries were screened against two subtypes of Influenza Hemagluttinin producing 4 total datasets (Phillips et al., 2021). The third dataset, referred to as G6, assesses all possible single–amino acid substitutions to the variable region of antibody G6.31, totaling 4,275 data points of binding with VEGF-A (Koenig et al., 2017). Due to experimental structures being unavailable for each antibody-antigen pair, homologous protein structures were used for some predictions, following Shanker et. al (Shanker et al., 2024). For the CR9114 dataset, the input structure used is CR9114 bound to the H5 antigen, despite being used to predict binding to H1 and H3 antigen subtypes. Additionally, for the CR6261 dataset, the input structure for predicting binding with both H1 and H9 subtypes, is actually a structure of CR6261 complexed with H1.

*Table 8.* Sidechain packing performance comparison across CASP datasets, measured by mean absolute error (MAE) of each chi angle. Bold values indicate the best results. Underlined values indicate second best results. Asterisks denote that the training dataset does not explicitly hold out CASP proteins and homologues. Results for Rosetta, DiffPack (Zhang et al., 2024), AttnPacker and AttnPacker-pp (McPartlon & Xu, 2023) are from Lee and Kim (Lee & Kim, 2024). All other models were evaluated in this work.

| Dataset | Method | Angle MAE (°) ↓ | | | | RMSD (Å) |
| | | $\chi_1$ | $\chi_2$ | $\chi_3$ | $\chi_4$ | All |
|---|---|---|---|---|---|---|
| CASP13 | Rosetta | 24.84 | 30.96 | 45.35 | 58.28 | 0.822 |
| | DiffPack | 22.14 | 28.80 | 45.20 | 52.28 | 0.789 |
| | AttnPacker | 17.82 | 33.41 | 67.31 | **48.89** | 0.745 |
| | AttnPacker-pp | 16.33 | 26.00 | 51.18 | 49.40 | 0.676 |
| | 3DCNN* | 24.74 | 33.03 | 46.96 | 56.73 | 0.787 |
| | ChromaDesign* | 17.61 | 26.14 | 43.76 | 55.05 | 0.677 |
| | LigandMPNN* | 16.19 | **23.96** | **38.46** | 49.02 | 0.680 |
| | FlowPacker | 16.58 | 25.04 | 43.09 | 51.41 | 0.683 |
| | FAMPNN (0.3Å) | 20.61 | 26.01 | 45.31 | 50.33 | 0.667 |
| | FAMPNN (0.0Å) | **14.94** | 25.07 | 41.05 | **48.89** | **0.579** |
| CASP14 | Rosetta | 32.32 | 35.47 | 49.19 | 54.27 | 1.001 |
| | DiffPack | 31.00 | 34.43 | 51.72 | 57.50 | 0.994 |
| | AttnPacker | 27.29 | 39.26 | 67.94 | 49.99 | 0.955 |
| | AttnPacker-pp | 26.06 | 32.75 | 55.06 | 50.59 | 0.900 |
| | 3DCNN* | 31.56 | 36.50 | 51.29 | 60.57 | 0.952 |
| | ChromaDesign* | 25.24 | 31.47 | 47.27 | 55.94 | 0.851 |
| | LigandMPNN* | 22.06 | **28.98** | **43.52** | **47.36** | 0.825 |
| | FlowPacker | 23.09 | 30.26 | 47.46 | 51.14 | 0.838 |
| | FAMPNN (0.3Å) | 26.02 | 31.24 | 47.78 | 53.08 | 0.821 |
| | FAMPNN (0.0Å) | **21.53** | 30.72 | 47.96 | 52.21 | **0.745** |
| CASP15 | Rosetta | 32.37 | 35.22 | 45.29 | 58.92 | 0.938 |
| | DiffPack | 31.86 | 34.46 | 48.06 | 61.01 | 0.921 |
| | AttnPacker | 28.16 | 41.90 | 69.90 | 53.22 | 0.925 |
| | AttnPacker-pp | 26.80 | 32.74 | 56.78 | 53.98 | 0.851 |
| | 3DCNN* | 32.12 | 36.53 | 50.43 | 62.37 | 0.897 |
| | ChromaDesign* | 26.34 | 31.72 | 44.44 | 59.83 | 0.810 |
| | LigandMPNN* | 23.32 | **28.97** | **41.16** | **51.87** | 0.788 |
| | FlowPacker | 23.63 | 29.26 | 42.97 | 53.35 | 0.765 |
| | FAMPNN (0.3Å) | 27.96 | 31.19 | 45.33 | 54.92 | 0.785 |
| | FAMPNN (0.0Å) | **21.75** | 29.49 | 43.07 | 55.41 | **0.690** |

