# OpenReview forum: "Sidechain conditioning and modeling for full-atom protein sequence design with FAMPNN"
_ICML.cc/2025/Conference — ICML 2025 poster_

### Official Review · Reviewer_mv2D · 2025-02-28

**Overall Recommendation:** 4

**Summary:**

The paper presents a new model FAMPNN for fixed-backbone protein sequence design that models both sequence and sidechains. FAMPNN addresses the limitations of existing methods that rely solely on backbone and sequence identity. The authors demonstrate that FAMPNN improves sequence recovery, achieves state-of-the-art sidechain packing, and can be used for zero-shot prediction of binding and stability.

**Claims And Evidence:**

yes

**Essential References Not Discussed:**

the authors could expand more on recent all-atom approaches for protein generation (e.g. Protpardelle, ProteinGenerator etc.) and co-folding (AF3).

**Experimental Designs Or Analyses:**

yes

**Methods And Evaluation Criteria:**

yes

**Other Comments Or Suggestions:**

/

**Other Strengths And Weaknesses:**

Strengths:
* The paper is well written and handles an interesting and challenging problem of joint sequence and sidechain conformation prediction. The proposed model is novel and well motivated.
* The authors provide extensive experimental results, demonstrating the effectiveness of FAMPNN across multiple benchmarks (zero-shot prediction of protein fitness, side chain packing, sequence design). The inclusion of the sidechain confidence module adds interpretability to the model's predictions.

Weakness:
* The authors could include a comparison in sampling time of FAMPNN compared to backbone-only models like ProteinMPNN to measure the computational cost of predicting sidechains.

**Questions For Authors:**

* What is the standard deviation in the predictions ? Given that the process is the outcome of diffusion process, there might be some variance in the prediction.
* Section 5.3; is it possible to compare on Figure 5.a and 5.c. these results with a supervised baseline to provide a reference point ?
* Can the authors comment why in Table 5. ProteinMPNN is working better for higher lengths (400, 500); is this due to the error in side chain coordinate predictions ?
* more generally, why is there a need for predicting sequence in the model given that the full atom representation contains implicitely information on the sequence ?
* could the authors comment on the application of FAMPNN to multi chain sequence design problems ? I gather from Section 4.3 that the model can be applied to complexes.

**Relation To Broader Scientific Literature:**

By integrating sequence and sidechain modeling in a single model, the paper extends prior work (e.g. ProteinMPNN which focuses on predicting protein sequences based on backbone structure without sidechain modelling) for improve protein design.

**Theoretical Claims:**

no theoretical claims

---

> ### Author Rebuttal · Authors · 2025-04-01
>
> We thank the reviewer for their insightful comments and questions. Below, we address these questions in detail:
>
> > FAMPNN sampling time comparisons
>
> Please refer to the last section of the response to reviewer GPV6
>
> > Standard deviation in the predictions
>
> Please refer to the first section of the response to reviewer MU8A
>
> > Supervised baselines for Figure 5
>
> This is an excellent suggestion, for **5c** there are baselines available with formal test splits for stability datasets on which we can evaluate FAMPNN to compare to supervised methods. Supervised baselines and formal test splits do not exist for **5a**, but we can run our own baseline (eg. one-hot-encoding of sequence with a ML-based regressors). If accepted, we will include these additional results.
>
> > ProteinMPNN self-consistency on longer lengths
>
> This is an interesting hypothesis that we also explored. We believe that side-chain prediction error may indeed be compounding for longer-length proteins. At each sampling step, we condition only on previously-predicted side chains with predicted error < 0.3Å, and notice a larger performance degradation without this filtering.
>
> > Why explictily predict sequence?
>
> We agree that given a full-atom representation, it is trivial to predict the sequence as it is already defined by the atom composition. In early versions of the model, we experimented with removing explicit sequence prediction and calling sequence from the implicit full-atom predictions, but we found this to not work as well in practice. We believe that the cross-entropy loss objective is important for the model to learn to predict high-quality sequences. Additionally, in addition to careful tuning of our masking schedule, and occasionally completely dropping all side-chain coordinates, FAMPNN is able to predict sequences with no full-atom representation, allowing use in situations such as sequence design for de novo backbones.
> > Multi-chain sequence design with FAMPNN
>
> Please refer to the rebuttal to reviewer GPV6 for details on the training of the PDB model and the respective dataset. Regarding multi-chain design, we showcase the capability of the PDB model for multi-chain design in Figure **6b**, where we demonstrate the utility of full-atom context for higher sequence recovery in protein-protein interfaces. Here, FAMPNN achieves higher sequence recovery, and higher sequence recovery scaling with context than LigandMPNN. Additionally, we note FAMPNN performs state-of-the-art in binding-affinity fitness evaluation in large multi-chain protein-protein complexes greater 3000 residues (SKEMPIv2 dataset), which are shown in figures **5a** and **5c**. Together these results demonstrate robust utility of FAMPNN in multi-chain design.

---

### Official Review · Reviewer_MU8A · 2025-03-13

**Overall Recommendation:** 3

**Summary:**

This paper introduces FAMPNN (Full-Atom MPNN), a model that explicitly incorporates sidechain conformation modeling for fixed-backbone protein sequence design. While existing deep learning methods implicitly reason about sidechain interactions based solely on backbone geometry and amino acid sequence, FAMPNN jointly models both sequence identity (discrete) and sidechain conformation (continuous) for each residue using a combined categorical cross-entropy and diffusion loss objective, respectively.

Built on a hybrid MPNN-GVP architecture that conditions on both backbone and available sidechain information, FAMPNN employs an iterative sampling strategy to efficiently generate samples from the joint distribution of sequence and structure. The model achieves competitive sequence recovery on CATH 4.2 and strong self-consistency results compared to state-of-the-art methods like ProteinMPNN when evaluated on de novo backbones. FAMPNN also demonstrates superior sidechain packing accuracy on CASP13/14/15 datasets and provides per-atom predicted sidechain error estimates that strongly rank-correlates with true errors. Additionally, the authors show FAMPNN's effectiveness for unsupervised fitness prediction on experimental datasets for antibody-antigen binding, protein-protein binding affinity, and protein stability.

Through comprehensive analysis, the authors demonstrate that increasing sidechain context (both sequence identity and conformation) leads to better model performance. Their ablation studies reveal that a similarly sized model without sidechain context performs worse for protein-protein binding affinity prediction, though it closely matches FAMPNN's performance for antibody-antigen binding affinity and protein stability prediction.

**Claims And Evidence:**

- **Claim**: pSCE serves as an effective confidence metric for sidechain packing
  While this claim is supported by high Spearman correlation in Figure 3(a), there is miscalibration between predicted and true sidechain error. In the per-residue setting, the maximum pSCE is around 2.0 in contrast to the maximum true error of 4.0. This limits its utility as an absolute error predictor, though it remains useful for ranking/relative confidence.

- **Claim**: FAMPNN outperforms unsupervised methods for antibody-antigen binding affinity and protein stability
  This claim is supported by Figure 4(a) and 4(c) when FAMPNN is compared to other structure-conditioned models. The authors should also provide performance comparison with leading sequence-only foundation models or qualify in text that the comparison is limited to structure-conditioned models. It's also unclear why the comparison doesn't include other state-of-the-art methods like ESM3.

- **Claim**: Increasing context leads to better sequence packing accuracy
  This is more or less true but the relationship between context and packing accuracy is not monotic, strictly speaking. There is a nominal increase in RMSD around 40% context for partial sidechain context and a similar increase around 50% context for partial sequence context which should be clarified or explained in the text.

**Essential References Not Discussed:**

The most notable exclusion from the paper is the Chroma generative model (https://www.nature.com/articles/s41586-023-06728-8; Nature 2023) that addresses the problem of incorporating sidechain conformation into fixed-backbone sequence design amongst other things and has overlapping ideas.

**Experimental Designs Or Analyses:**

**Strengths:**
- Comprehensive benchmarking and analysis of sequence recovery, self-consistency, sidechain packing, zero-shot fitness prediction, and the impact of available sidechain context on performance.
- Selection of a wide range of appropriate baseline models for comparison.
- Rigorous ablation study is performed to investigate the primary claim of the paper about the utility of sidechain context.
- The authors ensure there is no data leakage in the evaluation by holding out validation/test data from training and use updated splits for datasets like Skempi to ensure fair comparison.

**Weaknesses:**
- The authors do not provide interval estimates for performance metrics which makes it harder to evaluate significance.
- The selection of benchmark models varies across evaluations without clear justification. For instance, ESM3 is part of the self-consistency evaluation but not the zero-shot fitness prediction evaluation.
- A notable exclusion from the paper is the Chroma generative model (https://www.nature.com/articles/s41586-023-06728-8) which also addresses the problem of incorporating sidechain conformation into fixed-backbone sequence design using similar architectural components. It would have been nice to see performance comparison with this model.
- For antibody-antigen binding datasets, the authors note they used homologous structures when exact experimental structures weren't available (e.g., using CR9114 bound to H5 to predict binding to H1 and H3 subtypes). This potential mismatch might introduce errors in evaluation.

**Methods And Evaluation Criteria:**

**Strengths:**
- The formulation of the training objective considering both discrete sequence tokens and continuous sidechain coordinates is appropriate for the problem at hand.
- The proposed method re-uses existing components (MPNN, GVP) that have been proven to be effective.
- The selected datasets and metrics are well-suited to the problem being addressed. In particular, evaluating sequence recovery and self consistency on CATH 4.2 is a standard practice in the field.
- A diverse set of datasets covering antibody-antigen binding, protein-protein binding affinity, and protein stability are used for zero shot evaluation of fitness prediction.

**Weaknesses:**
- Sequence recovery aggregates binary decisions for correct sequence identity without taking into account the precise confidence of the model for the correct residue. Perplexity addresses this shortcoming, however, it is not included in the evaluation.
- ProteinGym (https://proteingym.org/) is a standard benchmark in the field for zero-shot fitness prediction that is not included in the evaluation making it difficult to compare performance with other methods not part of authors' evaluation.

**Other Comments Or Suggestions:**

Typos:
- Missing year of publication for Akpinaroglu et al. in the citation in text. (e.g. page 1, first column, line 49)
- The citation for RFdiffusion is missing in text.
- Page 7, second column, lines 341-362: The references to Figure 5 sub-figures in the paragraph appear incorrect.
- Page 23, line 1251: "provided the the"

Suggestions:
- Please mention the number of unmasking steps used for generating Figure 2a.
- Please consider remaking Figure 3a to include Pearson correlation as well as marginal distribution of true and predicted sidechain errors.
- Please mention the size of different datasets used for evaluation in section 5.3 in the main text and the kind of structure (predicted, crystal, etc.) available for each.
- Please consider remaking Figure 6a to share the y-axis range for the two subplots.

**Other Strengths And Weaknesses:**

The paper is well written and easy to follow. The authors provide the key implementation details, descriptions of datasets along with preprocessing details, and describe the evaluation criteria in detail.

**Questions For Authors:**

1. Could the authors clarify the novelty of their work in the context of Chroma (https://www.nature.com/articles/s41586-023-06728-8; Nature 2023) and provide a more detailed performance comparison? This would help in understanding the unique contributions of FAMPNN and allow for a more accurate assessment of the paper's impact.

**Relation To Broader Scientific Literature:**

The primary contribution of FAMPNN is the explicit modeling of sidechain conformation during fixed-backbone sequence design. This is a known limitation of existing methods widely used in the field such as ESM-IF1, ProteinMPNN, etc. The authors show that addressing this limitation leads to improved performance along various axes. However, the novelty of the work is unclear in the context of Chroma (https://www.nature.com/articles/s41586-023-06728-8) which also addresses the problem of incorporating sidechain conformation into fixed-backbone sequence design and has overlapping ideas.

**Theoretical Claims:**

No theoretical claims are made in this paper.

---

> ### Author Rebuttal · Authors · 2025-04-01
>
> We appreciate the reviewer’s comprehensive evaluation and suggestions to improve clarity and benchmark comparisons in our paper. Below, we provide detailed responses and clarifications:
>
> > Inclusion of perplexity in addition to sequence recovery for evaluation
>
> This is a great suggestion. With 1 step recovery, FAMPNN achieves a perplexity of 4.99, and we will update this with the perplexities of all methods in Table 1 in a revised version.
>
> > Evaluation on ProteinGym
>
> We agree with the reviewers point and will include ProteinGym in our evaluations if accepted. We would also like to note that many of the assays included in our fitness evaluations make up a great amount (but of course not all) of ProteinGym data (eg. Megascale, SKEMPIv2).
>
> > Interval estimates
>
> We thank the reviewer for pointing this out and believe it would be a valuable addition. We intend to add interval estimates for performance metrics in the final version if accepted.
>
> > Expected monotonic relationship between context and packing accuracy
>
> We believe that the slight deviations from this trend are due to random chance, and we plan to update this plot with error bars as well as clarify this claim in the text.
>
> > Selection of benchmark models
>
> In general, we chose to benchmark models that are commonly used for their respective tasks. We agree that ESM3 should be included in the zero-shot fitness prediction evaluations, and we will include this in the final version if accepted.
>
> > Novelty of our work in the context of Chroma
>
> As correctly pointed out, Chroma does in fact have the capability to return an all-atom structure given an input backbone via its ChromaDesign module. However, the sequence design module does not have the ability to encode a full-atom protein structure and therefore cannot design sequences conditioned on neighboring sidechain context. As a consequence, this means that **Chroma's sequence predictions are unable to leverage either experimentally determined sidechain conformations or previously predicted sidechains**. By contrast, FAMPNN **explicitly conditions on sidechain context**, which allows users to provide known sidechain conformations as conditioning input during sequence generation. This allows for more fine-grained control. Despite this, we very much agree it would be good to discuss Chroma and benchmark ChromaDesign along with other methods. We will provide updates in a revised version and have included an extension to Table 5 below:
>
> | Method        | Length 100     | Length 200     | Length 300     | Length 400     | Length 500     |
> |---------------|----------------|----------------|----------------|----------------|----------------|
> |               | scTM \| pLDDT  | scTM \| pLDDT  | scTM \| pLDDT  | scTM \| pLDDT  | scTM \| pLDDT  |
> | FAMPNN (0.3Å) | 0.968 \| 93.00 | 0.967 \| 91.27 | 0.938 \| 83.45 | 0.760 \| 74.73 | 0.545 \| 61.73 |
> | FAMPNN (0.0Å) | 0.896 \| 88.99 | 0.890 \| 81.87 | 0.703 \| 67.86 | 0.602 \| 63.80 | 0.471 \| 55.49 |
> | Chroma        | 0.940 \| 90.94 | 0.949 \| 88.04 | 0.946 \| 86.73 | 0.914 \| 80.02 | 0.751 \| 71.50 |
>
> We also provide an extension to Table 2 for sidechain packing evaluation below, noting that Chroma has not been trained on a dataset to explicitly hold out CASP homologues:
>
> | Dataset | Method        | Atom RMSD             |
> |---------|---------------|-----------------------|
> |         |               | All / Core / Surface  |
> | CASP13  | FAMPNN (0.3Å) | 0.667 / 0.362 / 0.775 |
> |         | FAMPNN (0.0Å) | 0.579 / 0.345 / 0.659 |
> |         | Chroma*        | 0.677 / 0.392 / 0.770 |
> | CASP14  | FAMPNN (0.3Å) | 0.821 / 0.534 / 0.937 |
> |         | FAMPNN (0.0Å) | 0.745 / 0.430 / 0.858 |
> |         | Chroma*        | 0.851 / 0.550 / 0.964 |
> | CASP15  | FAMPNN (0.3Å) | 0.785 / 0.417 / 0.888 |
> |         | FAMPNN (0.0Å) | 0.690 / 0.350 / 0.789 |
> |         | Chroma*        | 0.810 / 0.434 / 0.917 |
>
>
> > Potential mismatch for antibody-antigen binding dataset evaluation
>
> Yes this is true. As described in Shanker et. al, experimental structures matching the exact sequence are not available for each assay, however as the sequences of the different HA variants (H1,3,9 etc.) are quite close, models are still able to perform well using the backbone of the homologous proteins. While this can bring about errors, and we do acknowledge this, this is an extremely common occurrence in practical protein design, where homologous structures must be used due to a lack of experimental structure (homology modelling). Thus, we see this as an opportunity to showcase the model's performance in a structurally data limited situation which is even more prudent to demonstrate given our method uses “more” structure than its backbone-only counterparts.
>
> > Other comments or suggestions
>
> We thank the reviewer for suggestions on improving the clarity of our figures and will make these changes if accepted.

---

### Official Review · Reviewer_CPZt · 2025-03-14

**Overall Recommendation:** 4

**Summary:**

This paper presents FAMPNN, an iterative inverse folding algorithm capable of co-generating sidechain conformations and sequences.
Such design allows the model to condition on the currently known sidechain atoms in addition to the fixed backbone and sequence.
FAMPNN models the per-residue sequence type and sidechain structure with a combined cross-entropy and diffusion loss objective.
Experimental results show that FAMPNN can achieve promising results in full-atom sequence design, sidechain packing, and full-atom conditioned protein fitness evaluation. They also validate the effectiveness of the sidechain context.

## update after rebuttal
I've read the authors' replies and other reviewers' comments. I will keep my positive score.

**Claims And Evidence:**

Yes

**Essential References Not Discussed:**

No

**Experimental Designs Or Analyses:**

Yes, the experimental designs are sound as they benchmark the proposed method on 3 related tasks and also perform an additional ablation study.

**Methods And Evaluation Criteria:**

Yes

**Other Comments Or Suggestions:**

NO

**Other Strengths And Weaknesses:**

Strengths
1. sound experiment design and results
2. They show that the FAMPNN can perform protein fitness evaluation quite well. We may scale up the training data (with some af2 distillation data) and model size to get even better results.

**Questions For Authors:**

1. In section 4.2.1, the authors mention the definition of the ghost atom. However, it is not clear how they encode the ghost atoms for masked tokens.

2. FAMPNN adopts an MLM framework for inverse folding instead of random permutation AR or diffusion. I'm curious whether the MLM framework is optimal for the inverse folding task. Have you ever tried using a random permutation AR (proteinmpnn) for FAMPNN?

**Relation To Broader Scientific Literature:**

The key contributions of the paper are related to next-token modeling for continuous data.

**Theoretical Claims:**

NA.

---

> ### Author Rebuttal · Authors · 2025-04-01
>
> We'd like to thank the review for their positive evaluation and interest in the methodological choices behind our model. Below, we clarify these points:
>
> > Ghost atoms for masked tokens
>
> To encode the ghost atom for masked tokens, we set them at the position of the central CA atom by default. Because the model receives a mask sequence token for this position, it is able to understand that these ghost atoms refer to a masked position.
>
> > MLM vs. AR
>
> In early versions of the model, we experimented with both AR and MLM, finding that they perform similarly in terms of both sequence recovery and self-consistency. We noticed that high-quality sequences could be predicted in relatively few steps, so viewing the generative process through an MLM framework could allow for faster inference. We also chose MLM because it gives us more flexibility in choosing train-time masking schedules, allowing the model to prioritize learning on certain masking levels.

---

### Official Review · Reviewer_GPV6 · 2025-03-22

**Overall Recommendation:** 3

**Summary:**

The paper introduces FAMPNN for protein sequence design that explicitly models both the sequence identity and sidechain conformation of each residue. Unlike existing methods that rely solely on backbone geometry, FAMPNN uses a combined categorical cross-entropy and diffusion loss objective to jointly learn the distribution of amino acid identities and sidechain conformations. The authors demonstrate that this approach improves sequence recovery and achieves state-of-the-art sidechain packing accuracy. Additionally, FAMPNN shows promise in practical applications, such as zero-shot prediction of protein stability and binding affinity.

**Claims And Evidence:**

yes.

**Essential References Not Discussed:**

n/a

**Experimental Designs Or Analyses:**

yes.

**Methods And Evaluation Criteria:**

yes.

**Other Comments Or Suggestions:**

n/a

**Other Strengths And Weaknesses:**

**Strengths**

1. The explicit modeling of sidechain conformations during sequence design is a significant advancement over existing methods that only consider backbone geometry. This is a clear improvement, as sidechain interactions are crucial for protein stability and function.
2. The use of a diffusion loss for sidechain conformation prediction is well-justified. It allows the model to handle continuous data (sidechain coordinates) effectively.
3. The paper provides a thorough evaluation of FAMPNN across multiple benchmarks, including sequence recovery, sidechain packing, and protein fitness prediction. The results are strong. The inclusion of zero-shot prediction tasks (e.g., protein stability and binding affinity) is particularly compelling.

**Weaknesses**
1. In terms of sequence recovery, Table 1 lacks of recent and stronger baselines. This omission weakens the claim of state-of-the-art performance. Include 2023–2024 methods or clarify why they were excluded.
2. High sequence recovery ≠ good design. The field prioritizes novel, stable sequences that fold into the target structure, not just matching native sequences. The lack of diversity and novelty metrics leaves it unclear if FAMPNN avoids overfitting. This is not only a practical desire for protein design but also an important measure about how and what your model learns, capable of generalization or simply through memorization. Authors however lack of related results and discussions.
3. Training/inference costs (e.g., GPU hours, memory) are not quantified, raising concerns for scaling to large proteins or multi-chain complexes.

**Questions For Authors:**

n/a

**Relation To Broader Scientific Literature:**

n/a

**Theoretical Claims:**

no theoretical contributions.

---

> ### Author Rebuttal · Authors · 2025-03-31
>
> We thank the reviewer for their suggestions to strengthen our baselines and evaluations. Below, we address the raised concerns:
>
> > Inclusion of more recent sequence recovery baselines
>
> We thank the reviewer for pointing out this omission. To this end, we report Frame2Seq (Dec. 2023), a structure-conditioned masked protein language model achieve 46.53% sequence recovery as a single model, and 49.11% as an ensemble of three models on the CATH 4.2 test set with a single step (Akpinaroglu et. al), while FAMPNN achieves a 49.66% and 50% recovery as a single model with  single and 5-step sampling respectively. Second, some other baselines could not be included due to differences in training data: e.g. certain methods do not train on CATH and elect to train on non-standardized versions of the PDB, thus we cannot compare sequence recovery on a common test set. Third, certain recent baselines for sequence recovery, particularly hybrid models which utilize pre-trained protein language models such as ESM2 (e.g. LM-Design, KW-Design), may have data-leakage issues with the CATH 4.2 test set, as pretrained protein language models likely have trained on sequences in the CATH test set. We will include a discussion of this in the final version of the paper if accepted. Finally, in this paper, we note that we do not claim state-of-the-art performance in terms of sequence recovery, but rather that sidechain packing and sequence design are synergistic tasks to learn, where we can achieve self-consistency competitive with ProteinMPNN while achieving state-of-the-art packing.
>
> > High sequence recovery ≠ good design
>
> We completely agree that high sequence recovery does not equal good design. To this end we use the de novo benchmark, popularized by ProteinBench (Ye et. al), as a way of measuring the ability to recapitulate structures on structures that are structurally distinct and diverse from those found in nature and report self-consistency to capture if designs “fold into the target structure” as you mention. As these proteins are not found in nature, we hope this demonstrates generalization beyond memorization. In fact, we find very low average pairwise similarity between sequences generated by FAMPNN on this benchmark with **[13.4%, 9.1%, 8.4%, 7.9%, 7.8%]** average pairwise similarity for de novo backbones of length 100, 200, 300, 400, and 500 respectively. We will conduct a more robust benchmark of diversity on de novo backbones and compare to other methods in the final version of the paper if accepted.
>
> > Training and inference costs
>
> We completely agree that compute costs are important for a useful protein design tool. Our CATH and PDB models are competitive with state-of-the-art models at **8.6M** and **10.7M** parameters respectively, within magnitude of models such as ProteinMPNN (1.9M), Frame2Seq (10M), and PiFold (6.6M). Notably, our model is much smaller compared to models such as ESM-IF (142M), LM-Design (664M), and ESM-3 (1.4B). For training, we mention in Section B.2 that we trained our CATH model for **~8 hours** on a single H100 GPU with 80GB of memory. The PDB model was trained with 4 H100s for 72 hours, but in practice we notice that the model can reach similar performance in **~24 hours** (96 GPU hours). Regarding example size, training of our PDB dataset was conducted with examples cropped or padded to a large fixed size of 1024 residues, with 82.91% of examples being multi-chain complexes. Finally, regarding inference costs, when evaluating our CATH model on the test-set, we achieve a single-step sampling speed of **0.03s** and **0.11s** per sample on a single H100 GPU. Additionally, we are enthusiastic to add a detailed comparison of inference costs to other methods to the final version of the paper. We appreciate this point being, as we have found that because we use a MLM procedure, FAMPNN can sample equally high quality sequences with much fewer steps, making our method much faster for longer proteins than other methods, including ProteinMPNN.

---

### Decision · Program_Chairs · 2025-05-01

**Decision:**

Accept (poster)

**Comment:**

This paper presents a new protein sequence design method called FAMPNN (Full-Atom MPNN) by jointly generate both the sequence identity as well as the side chain conformations, which is very novel compared with existing sequence-only generation method. Experimental results show that the proposed approach improves both sequence recovery and also achieves state-of-the-art performance for sidechain packing. All reviewers agree that the paper presents a sound contribution to the field of protein sequence design, and hence the AC recommends the paper for acceptance.